# Vision-R1: Incentivizing Reasoning Capability in Multimodal Large Language Models

Wenxuan Huang[1,2*†]     Bohan Jia[1*]     Shaosheng Cao[3✉]

Zheyu Ye[3]     Fei Zhao[3]     Zhe Xu[3]     Yao Hu[3]     Shaohui Lin[1,4✉]

[1]East China Normal University     [2]The Chinese University of Hong Kong     [3]Xiaohongshu Inc.

[4]Key Laboratory of Advanced Theory and Application in Statistics and Data Science - MOE, China

`wxhuang0616@gmail.com` (Wenxuan Huang)

`shaohuilin007@gmail.com` (Shaohui Lin)

*: Equal Contribution   †: Project Leader   ✉: Corresponding Author

## Abstract

DeepSeek-R1-Zero has successfully demonstrated the emergence of reasoning capabilities in LLMs purely through Reinforcement Learning (RL). Inspired by this breakthrough, we explore how RL can be utilized to enhance the reasoning capability of MLLMs. However, direct training with RL struggles to activate complex reasoning capabilities such as questioning and reflection in MLLMs, due to the absence of substantial high-quality multimodal reasoning data. To address this issue, we propose the reasoning MLLM, *Vision-R1*, to improve multimodal reasoning capability. Specifically, we first construct a high-quality multimodal CoT dataset without human annotations by leveraging an existing MLLM and DeepSeek-R1 through modality bridging and data filtering to obtain a 200K multimodal CoT dataset, *Vision-R1-cold dataset*. It serves as cold-start initialization data for Vision-R1. To mitigate the optimization challenges caused by overthinking after cold start, we propose *Progressive Thinking Suppression Training (PTST)* strategy and employ Group Relative Policy Optimization (GRPO) with the hard formatting result reward function to gradually refine the model's ability to learn correct and complex reasoning processes on the multimodal math dataset. Comprehensive experiments show our model achieves an average improvement of ∼6% across various multimodal math reasoning benchmarks using only a 10K multimodal math data during RL training. Vision-R1-7B achieves a 73.5% accuracy on the widely used MathVista benchmark, which is only 0.4% lower than the leading reasoning model, OpenAI O1. Scaling up the amount of multimodal math data in the RL training, Vision-R1-32B and Vison-R1-72B achieves 76.4% and 78.2% MathVista benchmark scores, respectively. The datasets, weight and code will be released in: `https://github.com/Osilly/Vision-R1`.

## 1 Introduction

Enhancing the complex reasoning capability of Large Language Models (LLMs) remains one of the most challenging problems in Artificial Intelligence (AI), which is widely regarded as a critical pathway toward Artificial General Intelligence (AGI) (Jaech et al., 2024; DeepSeek-AI, 2025). Conventional inference paradigms typically rely on a simple "direct prediction" approach to generate concise final answers without explicit, structured intermediate reasoning steps, which often exhibits suboptimal performance on complex reasoning tasks (Jaech et al., 2024). OpenAI O1 (Jaech et al., 2024) was the first LLM with strong reasoning ability by using complex Chain-of-Thought (CoT) for training to achieve significant performance gains over prior LLMs. Meanwhile, various methods (Wei et al., 2022; Yao et al., 2023; Besta et al., 2024; Lightman et al., 2023; Uesato et al., 2022; Wang et al., 2023; Lai et al., 2024; Wan et al., 2024; Trinh et al., 2024; Xin et al., 2024; Muennighoff et al., 2025; Ye et al., 2025) have been explored to generate high-quality complex CoT reasoning and further advance the field by optimizing reasoning pathways.

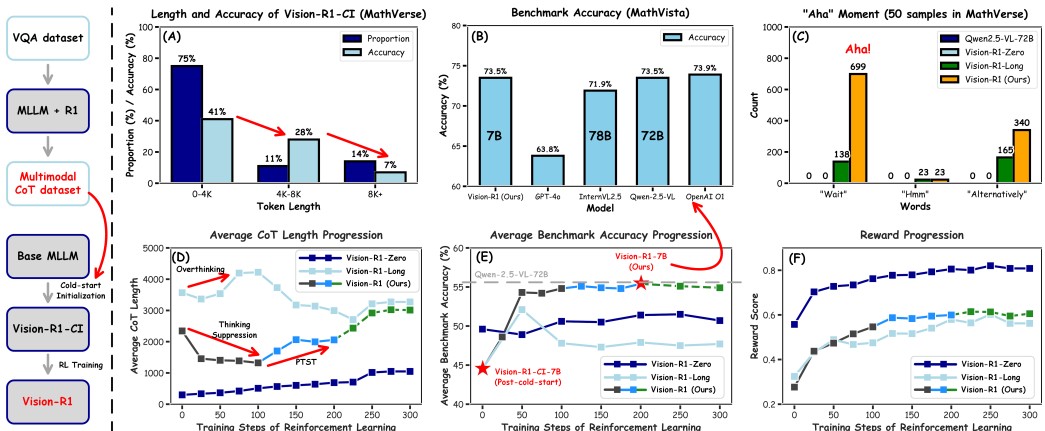

Figure 1: **Left panel:** Our Vision-R1 Pipeline. We first use the existing MLLM and DeepSeek-R1 to obtain a high-quantity Multimodal CoT dataset, which is used as the cold-start initialization data for the base MLLM to obtain the post-cold-start Vision-R1-CI, and then we perform the RL training on Vision-R1-CI to obtain the reasoning MLLM, Vision-R1. **Right panel:** We observe that directly applying RL to MLLMs fails to effectively incentivize strong reasoning capability (see (C) and (D)). Vision-R1-Zero, trained via RL without prior initialization, struggles to generalize from limited data (see (E), (F), notably, Vision-R1-Zero was applied in format reward function). Vision-R1-CI faces the Overthinking Optimization Problem, favoring shorter CoT reasoning, where correct reasoning processes mostly focus on the shorter CoT reasoning sequences (see (A)). During subsequent RL training, we observe a lengthening of reasoning steps but a decline in performance (see (D) and (E)), making optimization particularly challenging. For Vision-R1, it initially shortens CoT to refine the right thought process under RL training. PTST enables Vision-R1 to progressively acquire a more complex reasoning process (see (C), (D), and (E)) to improve the performance, such that our Vision-R1 with 7B parameters achieves comparable performance to the 70B+ strongest MLLMs (see (B)). Note that Vision-R1 used various colored lines to indicate the different stages in PTST.

In the field of Multimodal Large Language Models (MLLMs), recent works (Yao et al., 2024; Thawakar et al., 2025) have also explored the application of CoT reasoning. These approaches assume that MLLMs lack a structured reasoning process, achieving low performance on the tasks that require logical inference. To improve the reasoning capability of MLLMs, several methods (Xu et al., 2024; Yao et al., 2024) try to construct the datasets manually containing step-level reasoning processes and apply supervised fine-tuning (SFT) to reformat MLLMs' outputs. However, this manually designed "Formatted Reasoning MLLM" often results in "Pseudo-CoT" reasoning, which lacks essential cognitive processes commonly observed in human thoughts, such as questioning, reflection and inspecting (see Fig. 2, the data examples of "Pseudo-CoT" and the complex CoT). This limitation hinders their application on complex vision reasoning tasks. Thus, it is important to generate human-like, high-quality, complex CoT reasoning data for training MLLMs, enabling them to more effectively tackle intricate multimodal reasoning tasks.

Recently, DeepSeek-R1 (DeepSeek-AI, 2025) has successfully applied Reinforcement Learning (RL) to induce the self-emergence of complex cognitive reasoning ability in LLMs. This begs our rethinking: Can RL be utilized to incentivize the reasoning capability in MLLMs? To answer this question, we first follow the DeepSeek-R1-Zero paradigm (DeepSeek-AI, 2025), by directly using RL to improve the reasoning capability of MLLMs. Unfortunately, this direct RL training is challenged, as it struggles to effectively guide MLLMs generating complex CoT reasoning in absence of large-scale, high-quality multimodal data and prolonged training (see Fig. 1 (E) and (F)).

To address the above issue, we propose Vision-R1, a reasoning MLLM that integrates cold-start initialization with RL training. First, we construct a high-quality multimodal CoT dataset without requiring manual annotations. Specifically, we leverage an existing MLLM to generate "Pseudo-CoT" reasoning text from multimodal image-text pairs. This "Pseudo-CoT" reasoning explicitly incorporates both vision descriptions and structured step-level reasoning process, exposing more detailed vision information in a textual format. Next, we feed the enriched reasoning text back into the MLLM to obtain a description including necessary vision information. This process effectively implements "Modality Bridging", converting vision information to language. The resulting textual

descriptions are then passed to a text-only reasoning LLM, DeepSeek-R1, to extract high-quality CoT reasoning. Finally, the dataset is refined through rule-based data filtering, ultimately obtaining a dataset with 200K multimodal human-like complex CoT reasoning samples, which serves as the cold-start initialization dataset for Vision-R1.

Following the DeepSeek-R1 training pipeline, we need to apply Group Relative Policy Optimization (GRPO) (Shao et al., 2024; DeepSeek-AI, 2025) on a 10K multimodal math dataset to enhance the model's reasoning capability. However, as shown in Fig. 1 (A) and (D), we observe an overthinking phenomenon in the cold-start initialized MLLM, *i.e.*, the correct reasoning processes tend to be concentrated on shorter CoT reasoning sequences. This issue leads to the optimization problem in subsequent RL training. To address this challenge, we propose Progressive Thinking Suppression Training (PTST) alongside GRPO, which incorporates a hard-formatting result reward function. This approach encourages Vision-R1 to compress CoT reasoning steps early in the RL process, internalizing correct reasoning methods while progressively extending its reasoning duration over time to effectively tackle more complex problems.

Our main contributions can be summarized as follows:

1. We explore how to use R1-like RL for MLLMs and introduce Vision-R1, a reasoning MLLM that leverages cold-start initialization and RL training to incentivize reasoning capability. It is an early exploration that investigates the application of R1-like RL for enhancing reasoning capability in MLLMs and analyzes differences between direct RL training and the combined approach of cold-start initialization and RL training. We believe that our exploration can inspire new insights for the community.

2. A high-quality 200K multimodal CoT dataset without human annotations is constructed to serve as a cold-start initialization data for MLLMs. We leverage the proposed PTST to GRPO with hard-formatting result reward function, which effectively addresses the overthinking optimization problem in RL training. PTST enables Vision-R1 to progressively develop more complex reasoning processes while effectively guiding MLLMs toward enhanced reasoning capability.

3. Notably, despite having only 7B parameters, Vision-R1 achieves performance comparable to State-of-The-Art (SoTA) MLLMs with over 70B parameters in math reasoning tasks. Furthermore, the Vision-R1-32B and Vision-R1-72B models achieve an average accuracy improvement of around 10% compared to the base model on multiple multimodal math benchmarks.

## 2  RELATED WORK

### 2.1  LARGE LANGUAGE MODEL REASONING

As researchers have discovered that enabling LLMs to simulate human-like thought processes and perform stepwise reasoning can significantly enhance their performance on reasoning tasks (Jaech et al., 2024), extensive work has been dedicated to exploring LLM reasoning methods. These approaches typically rely on human design to format LLM outputs to follow specific steps, such as Chain-of-Thought (CoT) prompting methods (Wei et al., 2022), plan-based methods like Tree-of-Thought and Graph-of-Thought (Yao et al., 2023; Besta et al., 2024), process-based reward models (Lightman et al., 2023; Uesato et al., 2022; Wang et al., 2023; Lai et al., 2024), Monte Carlo Tree Search (MCTS) and Beam Search (Wan et al., 2024; Trinh et al., 2024; Xin et al., 2024), as well as constructing SFT datasets (Muennighoff et al., 2025; Ye et al., 2025).

A recent development, DeepSeek-R1 (DeepSeek-AI, 2025), demonstrates that applying large-scale Reinforcement Learning (RL) with formatting and result-only reward functions can guide LLMs toward self-emerging thought processes, producing human-like complex CoT reasoning and achieving significant advantages in complex reasoning tasks. This approach has shown immense potential in the field of Large Language Model Reasoning, however, its application to MLLMs remains an open area of inquiry.

### 2.2  MULTIMODAL LARGE LANGUAGE MODEL REASONING

MLLMs typically map inputs from other modalities to the textual modality, which are then processed by LLMs. This approach has been proven to exhibit superior performance in a range of vision

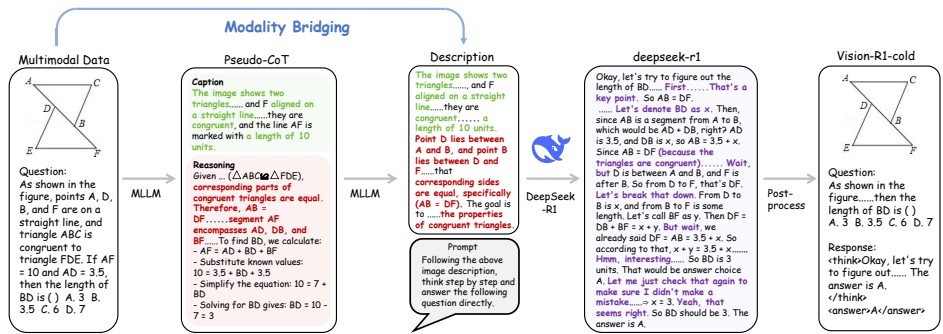

Figure 2: The overall data generation pipeline incorporating our Modality Bridging method. The multimodal data is first sent to MLLMs to obtain a "Pseudo-CoT" consisting of a caption and reasoning process, which serves as the input of MLLMs along with the original image-question pair to produce detailed descriptions. Through this modality bridging approach, the textual descriptions provide DeepSeek-R1 with holistic information that facilitates the generation of high-quality CoT processes, which are post-processed and integrated with the original data to create the final Vision-R1-cold dataset.

understanding tasks (Liu et al., 2024b;a; Bai et al., 2023; Zhu et al., 2023; Huang et al., 2024; Zhao et al., 2024b;a). Inspired by advancements in LLM reasoning, many studies have also sought to enhance the reasoning capability of MLLMs. For instance, efforts have been made to employ CoT prompting (Zhang et al., 2024b; Mitra et al., 2024; Luan et al., 2024) and to construct SFT datasets that include step-level reasoning (Yao et al., 2024; Thawakar et al., 2025). However, the CoT generated by these methods often lacks natural human cognitive processes, such as questioning, reflection, and inspecting, which limits their effectiveness in solving complex reasoning tasks. In contrast, Vision-R1 distinguishes itself by combining cold-start initialization with RL training to acquire high-quality, complex CoT reasoning capability.

## 3 METHOD

### 3.1 CAN ONLY RL INCENTIVIZE REASONING CAPABILITY IN MLLMS?

Inspired by DeepSeek-R1-Zero (DeepSeek-AI, 2025), we aimed to directly use RL to guide models towards self-thought and the emergence of complex reasoning capability. To this end, we collected a dataset of 10K open-source math problems for RL training. Specifically, we followed the DeepSeek-R1-Zero pipeline, training a base MLLM using GRPO (Shao et al., 2024), with output format constraints dictated by the following system prompt:

```
A conversation between User and Assistant. ... first thinks
about the reasoning process ... provides the user with the
answer. The reasoning process and answer are enclosed within
<think> </think> and <answer> </answer> tags ...
```

For the reward function, we utilized both formatting and result rewards:

1. Formatting reward function: The model's output must adhere to the "`<think> </think><answer> </answer>`" format.

2. Result reward function: The model's generated final result must align with the ground truth.

We set the formatting and result reward ratio as $1:1$.

The model after purely RL training is named as Vision-R1-Zero. Unfortunately, as shown in Fig. 1 (D) and (E), directly applying RL to train MLLMs has proven challenging in stimulating MLLM's reasoning capability and producing lengthy, complex CoT, limiting the model's reasoning ability. Moreover, we observed that as training progressed over extended periods, the model gradually learned to use longer reasoning processes to solve hard problems, but this did not show significant performance improvement. Thus, we claim that *directly using RL to incentivize the reasoning capability of MLLMs for solving complex reasoning problems remains a challenging task, especially **under constraints of data quality and quantity, as well as computation resources***.

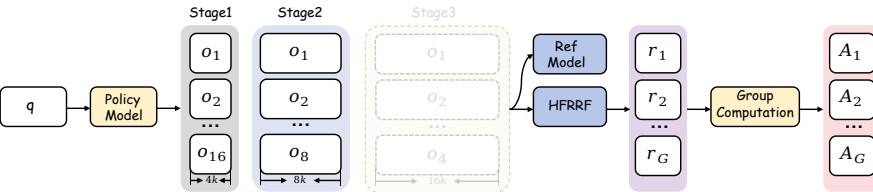

Figure 3: GRPO with our proposed PTST strategy. We progressively loosen the context length restrictions, increasing the length of reasoning process. Specifically, we set the reasoning length to 4K, 8K and 16K tokens for each stage, with corresponding group numbers of 16, 8 and 4 respectively. The reward function for GRPO is based on a hard formatting result reward function (HFRRF). The dotted line in the "Stage 3" indicates that the final version of Vision-R1 did not undergo the third stage of training.

## 3.2 OVERVIEW OF VISION-R1

In our above explorations, we observed that an RL-only approach struggles to guide MLLMs in generating human-like, complex CoT. Consequently, we explored an alternative strategy and introduced the reasoning MLLM, *Vision-R1*. This method begins with a cold-start using a multimodal CoT dataset, which initially teaches the base model to reason in a "human-like" manner. Subsequently, we apply RL to the cold-start initialized model Vision-R1-CI to guide it towards adopting the correct reasoning process, thereby incentivizing the reasoning capability in the final model Vision-R1.

In the following sections, we first describe our approach to create a high-quality, human annotation-free multimodal CoT dataset in Sec. 3.3.1, as well as the Overthinking Optimization Problem faced by post-cold-start MLLMs in Sec. 3.3.2. Then we discuss the RL training method, Progressive Thinking Suppression Training (PTST) in Sec. 3.4, to address the Overthinking Optimization Problem.

## 3.3 COLD-START INITIALIZATION

### 3.3.1 MODALITY BRIDGING TO OBTAIN HIGH-QUALITY MULTIMODAL CoT DATA

Many existing works (Xu et al., 2024; Yao et al., 2024) have attempted to construct multimodal reasoning datasets to enhance the reasoning capability of MLLMs. Prior efforts typically gathered CoT data which often lack the natural cognitive processes of questioning, and self-reflection. These datasets are typically constructed in a step-by-step form based on human heuristics. However, our goal is to collect a multimodal CoT dataset that encompasses complex cognitive processes to teach Vision-R1 to reason in a human-like, natural manner. Furthermore, DeepSeek-R1 has demonstrated the ability to generate CoT with natural cognitive processes and has proven to have strong reasoning capability. By using the high-quality CoT data it generates, which incorporates human-like cognitive self-reflection processes, we can train MLLMs to enhance their reasoning capability. However, being a "text-only" LLM, DeepSeek-R1 struggles to effectively process multimodal inputs to produce high-quality CoT.

To overcome these limitations, we utilize existing MLLMs alongside DeepSeek-R1 and propose a method named *Modality Bridging* to indirectly convert multimodal information into textual information, thereby capturing the complex cognitive processes of DeepSeek-R1. As the data generation pipeline that shown in Fig. 2, firstly, we follow prior works (Xu et al., 2024; Yao et al., 2024) by inputting an image-question-answer pair and a prompt into a MLLM to generate a "Pseudo-CoT" featuring both image description and reasoning processes. Subsequently, we concatenate the image-question pair with the "Pseudo-CoT" and a prompt, and then feed them into a MLLM to obtain a detailed image description. Below is the prompt template:

```
Given a image, a question:{question} and a thinking
process:{thinking process}, provide a detailed description
containing all the necessary details of the image to answer the
question correctly...
```

This process of generating "Pseudo-CoT" explicitly exposes more of the necessary vision details for reasoning in textual form, compared to pure image descriptions. This assists the MLLM in producing

Table 1: Comprehensive comparison with SoTA MLLMs (closed-source, open-source general/math/reasoning MLLMs) across diverse multimodal math benchmarks."Avg." denotes the average performance over all benchmarks. For MathVista benchmark, we have specifically compared all models on three sub-tasks that are highly related to mathematical reasoning: geometry reasoning (GEO), algebraic reasoning (ALG), geometry problem solving (GPS) and math word problems (MWP). "ALL" denotes the average score on MathVista benchmark. The best results are **bolded**.

| Model | Params. | MathVista | | | | | MathVerse | MM-Math | DynaMath | Avg. |
| --- | --- | --- | --- | --- | --- | --- | --- | --- | --- | --- |
| | | GEO | ALG | GPS | MWP | ALL | | | | |
| *Closed-Source MLLMs* | | | | | | | | | | |
| OpenAI O1 (Jaech et al., 2024) | – | – | – | – | – | 73.9† | – | – | – | – |
| GPT-4o (Hurst et al., 2024) | – | – | – | – | – | 63.8 | 37.6 | 31.8 | 64.9 | – |
| GPT-4V (OpenAI, 2023) | – | – | – | – | – | 49.9 | 39.4 | 23.1 | – | 37.5 |
| Claude-3.5 Sonnet (Anthropic, 2024) | – | – | – | – | – | 67.7 | 26.5 | – | 62.5 | – |
| *Open-Source General MLLMs* | | | | | | | | | | |
| Qwen2.5-VL-7B (Bai et al., 2025) | 7B | 66.9 | 68.7 | 66.8 | 76.9 | 68.1 | 46.7 | 34.1 | 50.7 | 49.9 |
| Qwen2.5-VL-32B (Bai et al., 2025) | 32B | 72.8 | 73.7 | 75.0 | 72.6 | 72.9 | 52.3 | 34.9 | 55.5 | 53.9 |
| Qwen2.5-VL-72B (Bai et al., 2025) | 72B | 77.8 | 77.9 | 78.8 | 74.7 | 73.5 | 51.3 | 45.6 | 61.2 | 57.9 |
| InternVL2.5-78B (Chen et al., 2024c) | 78B | 76.6 | 76.5 | 77.9 | 75.8 | 71.9 | 25.4 | 17.8 | 41.4 | 39.1 |
| *Open-Source Math MLLMs* | | | | | | | | | | |
| Math-LLaVA-13B (Shi et al., 2024) | 13B | 56.5 | 40.2 | 57.7 | 56.5 | 46.6 | 22.9 | – | – | – |
| Math-PUMA-Qwen2-7B (Zhuang et al., 2024) | 7B | 47.3 | – | 48.1 | – | 47.9 | 33.6 | – | – | – |
| Multimath-7B (Peng et al., 2024) | 7B | – | – | 66.8 | 61.8 | 50.0 | 26.9 | – | – | – |
| URSA-8B (Luo et al., 2025) | 8B | – | – | 79.3 | 75.3 | 59.8 | 45.7 | – | – | – |
| *Open-Source Reasoning MLLMs* | | | | | | | | | | |
| LLaVA-CoT-11B (Xu et al., 2024) | 11B | – | – | – | – | 54.8 | 20.3 | 16.5 | 34.6 | 31.6 |
| Mulberry-7B (Yao et al., 2024) | 7B | – | – | – | – | 63.1 | – | 23.7 | – | – |
| *Our Model* | | | | | | | | | | |
| Vision-R1-7B (Ours) | 7B | 80.3 | 79.0 | 83.2 | 80.6 | 73.5 | 52.4 | 40.2 | 56.3 | 55.6 |
| Δ (vs Qwen2.5-VL-7B) | | +13.4 | +10.3 | +16.4 | +3.7 | +5.4 | +5.7 | +6.1 | +5.6 | +5.7 |
| Vision-R1-32B* (Ours) | 32B | 85.8 | 82.6 | 88.0 | 78.5 | 76.4 | 62.1 | 55.3 | 65.6 | 64.9 |
| Δ (vs Qwen2.5-VL-32B) | | +13.0 | +8.9 | +13.0 | +5.9 | +4.7 | +9.8 | +20.4 | +10.1 | +11.0 |
| Vision-R1-72B* (Ours) | 72B | **88.3** | **86.8** | **89.4** | **79.6** | **78.2** | **63.2** | **59.3** | **66.4** | **66.8** |
| Δ (vs Qwen2.5-VL-72B) | | +10.5 | +8.9 | +10.6 | +4.9 | +4.7 | +11.9 | +13.7 | +5.2 | +8.9 |

† The result is collected from the official MathVista leaderboard (https://mathvista.github.io/#leaderboard).
* It uses additional data during RL training.

more detailed descriptions, thereby minimizing information loss during the conversion of original multimodal information to textual information (see Fig. 5).

At this stage, we cleverly bridge image information to textual information and feed it into DeepSeek-R1 to obtain high-quality CoT processes. We then retain reasoning processes from the generated multimodal CoT data that the final answer is aligned with ground truth and apply rule-based data filtering to remove logical inconsistent samples and replace some words for semantic coherence.

Finally, we pair the pure text CoT data generated by DeepSeek-R1 from the above process with the corresponding images, integrating into multimodal CoT data, named **Vision-R1-cold** dataset. This dataset is used for the cold-start initialization of Vision-R1. By acquiring CoT data in this manner, which closely mimics human cognitive behavior, the reasoning processes exhibit natural thinking.

### 3.3.2 OVERTHINKING OPTIMIZATION PROBLEM

After obtaining a multimodal CoT dataset, we conducted SFT on a pretrained MLLM (such as Qwen2.5-VL (Bai et al., 2025)) as the base MLLM for cold-start initialization. The MLLM after cold start initialization is named as Vision-R1-CI. At this stage, the base MLLM had learned the complex reasoning mode from DeepSeek-R1, however, this led to the Overthinking Optimization Problem, *i.e.*, Vision-R1-CI would engage in prolonged thought processes on certain problems, whereas the correct reasoning processes were typically concentrated in shorter cognitive chains.

As shown in Fig. 1 (D) and (E), this propensity for excessive incorrect reasoning significantly complicates the optimization of subsequent RL training. For instance, when the allowed thought length during RL training for Vision-R1-CI was directly extended to 16K, the model tended to generate longer answers to fulfill the demands of complex reasoning. However, this incorrect reasoning did not lead to performance improvements, thus presenting challenges in incentivizing reasoning capability in MLLMs. So, *it is crucial to guide the model to learn correct thinking in the early stages for the reasoning performance of MLLMs.*

Table 2: Comparison with SoTA MLLMs across diverse comprehensive multimodal benchmarks. The base MLLM is Llama-3.2-11B-V-Instruct (Dubey et al., 2024). The results indicate that our data significantly enhances the generalization capabilities of our model, leading to superior performance across all benchmarks.

| Method | General Benchmark | | | | Math Benchmark | | |
|---|---|---|---|---|---|---|---|
| | MMStar | ChartQA | $MME_{sum}$ | HallBench | MathVista | MathVerse | MM-Math |
| Llama-3.2-11B-V (Dubey et al., 2024) | 49.8 | 83.4 | 1787 | 40.3 | 48.6 | 8.4 | 4.1 |
| Mulberry-Llama-11B (Yao et al., 2024) | 58.5 | 83.5 | 2035 | 48.9 | 61.1 | – | 18.7 |
| LLaVA-Cot-11B (Xu et al., 2024) | 57.6 | 81.9 | 2137 | 47.8 | 54.8 | 20.3 | 16.5 |
| Vision-R1-LlamaV-CI-11B | 61.4 | 83.9 | 2190 | 49.5 | 62.7 | 27.1 | 26.1 |

## 3.4 PROGRESSIVE THINKING SUPPRESSION TRAINING

Inspired by the above phenomenon, we propose the Progressive Thinking Suppression Training (PTST) algorithm to initially suppress the length of reasoning during the early stages of RL training for Vision-R1, while guiding it towards the correct reasoning processes. As training progresses, we gradually relax these constraints, allowing Vision-R1 to autonomously learn to utilize longer CoT to address increasingly complex problems, thereby enhancing its reasoning capability. Specifically, we implement Group Relative Policy Optimization (GRPO) with hard formatting result rewards for the model's self-learning. Consider the standard GRPO approach, it samples a group of generated output set $\{o_1, o_2, \cdots, o_G\}$ for each question $q$ from policy model $\pi_{\theta_{old}}$. Then GRPO maximizes the following objective and optimizes the model $\pi_\theta$.

$$J_{\text{GRPO}}(\theta) = \mathbb{E}_{q \sim P(Q), \{o_i\}_{i=1}^G \sim \pi_{\theta_{\text{old}}}(O|q)}$$
$$\left[ \frac{1}{G} \sum_{i=1}^G \min\left( \frac{\pi_\theta(o_i \mid q)}{\pi_{\theta_{\text{old}}}(o_i \mid q)} A_i, \text{clip}\left( \frac{\pi_\theta(o_i \mid q)}{\pi_{\theta_{\text{old}}}(o_i \mid q)}, 1 - \varepsilon, 1 + \varepsilon \right) A_i \right) - \beta D_{\text{KL}}\left( \pi_\theta \,\|\, \pi_{\text{ref}} \right) \right], \quad (1)$$

where $\varepsilon$ and $\beta$ are the PPO clipping hyper-parameter and the coefficient controlling the Kullback–Leibler (KL) penalty (Shao et al., 2024; Schulman et al., 2017), respectively. We set $\varepsilon$=0.2 and $\beta$=1e-2 during training. $A_i = \frac{r_i - \text{mean}(\{r_1, r_2, ..., r_G\})}{\text{std}(\{r_1, r_2, ..., r_G\})}$ is the computed advantage using the group rewards $\{r_1, r_2, \cdots, r_G\}$ and $D_{KL}\left( \pi_\theta \,\|\, \pi_{\text{ref}} \right) = \frac{\pi_{\text{ref}}(o_i|q)}{\pi_\theta(o_i|q)} - \log\left( \frac{\pi_{\text{ref}}(o_i|q)}{\pi_\theta(o_i|q)} \right) - 1$ is the KL divergence.

As shown in Fig. 3, in our proposed PTST, we denote the total number of training stages by $S$, with each stage having its own sampling count $G_s$ and sequence length limit $L_s$. The output space for stage $s \in \{1, 2, \ldots, S\}$ is $O^{(s)} = \{ o : |o| \leq L_s \}$. The training objective for the $s$-th stage can be further reformulated based on Eq. 1 as:

$$J_{\text{GRPO}}^{(s),\text{w/PTST}}(\theta) = \mathbb{E}_{q \sim P(Q), \{o_i^{(s)}\}_{i=1}^{G_s} \sim \pi_{\theta_{\text{old}}}\left(O^{(s)}|q\right)}$$
$$\left[ \frac{1}{G_s} \sum_{i=1}^{G_s} \min\left( \frac{\pi_\theta\left(o_i^{(s)} \mid q\right)}{\pi_{\theta_{\text{old}}}\left(o_i^{(s)} \mid q\right)} A_i^{(s)}, \text{clip}\left( \frac{\pi_\theta\left(o_i^{(s)} \mid q\right)}{\pi_{\theta_{\text{old}}}\left(o_i^{(s)} \mid q\right)}, 1 - \varepsilon, 1 + \varepsilon \right) A_i^{(s)} \right) - \beta D_{\text{KL}}\left( \pi_\theta \,\|\, \pi_{\text{ref}} \right) \right], \quad (2)$$

where $A_i^{(s)}$ denotes the advantage estimate for the $i$-th sample in the training stage $s$, and $\pi_{\theta_{\text{old}}}$ denotes the policy model with an output length constraint of $O(s)$. We employ the hard formatting result reward function as the reward mechanism for GRPO, *i.e.*, the model receives a reward score of $r_i = 1$ only when both the formatting requirements and the correctness of the final answer are simultaneously satisfied; otherwise, it receives a score of $r_i = 0$. Moreover, we do not impose constraints using a system prompt, as Vision-R1-CI has already acquired robust formatting capability during the cold-start initialization.

By applying PTST, we compress the model's thought length in the early training stages to guide correct reasoning and gradually relax these constraints in later stages. As illustrated in Fig. 1, this progressive strategy enables Vision-R1 to generate more complex CoT and significantly enhances its reasoning capability. Notably, in practice, we observe that Vision-R1 achieves competitive performance by the end of the second training stage, leading us to select it as the final stage, *i.e.*, we set the parameters as $S = 2$, $L_s \in \{4K, 8K, 16K\}$, and $G_s \in \{16, 8, 4\}$, respectively.

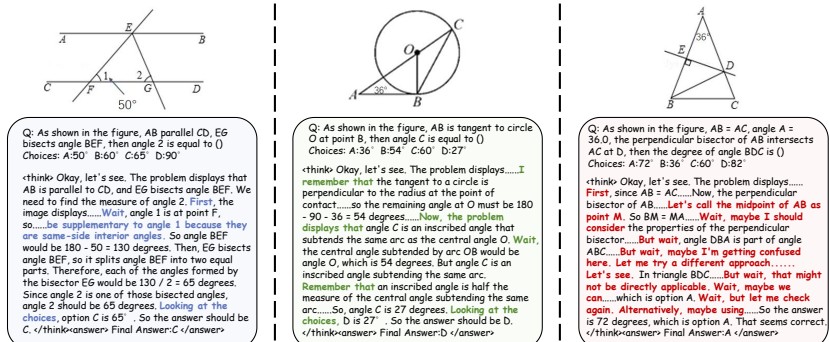

Figure 4: The output examples of Vision-R1-7B on MathVerse benchmark. Vision-R1-7B shows "human-like" questioning and self-reflective thought process when solving math reasoning problems, which is also called **"Aha moment"** in DeepSeek-R1's paper (DeepSeek-AI, 2025). More examples are provided in supplementary materials.

## 4 EXPERIMENTS

Please refer to Appendix A to obtain the detailed data and training settings. We provide the main results of Vision-R1 in Sec 4.1, and then we introduce the detailed ablation study in Sec 4.2.

### 4.1 MAIN RESULTS

As shown in Tab. 1, our proposed Vision-R1-7B achieves competitive results across multiple math reasoning benchmarks, even when compared to SoTA models with over 10 times the parameters of Vision-R1-7B. For instance, on the MathVista benchmark, Vision-R1-7B achieves a score of 73.5%, only 0.4% lower than OpenAI O1, the most widely recognized reasoning model.

Table 3: Evaluating the impact of Cold Start Initialization, GRPO, Progressive Thinking Suppression Training (PTST). "Avg. Len." denotes the average output token length on the 10K multimodal math dataset for RL training. "Avg. Acc." denotes the average performance (MathVista, MathVerse, MM-Math).

| Method | Cold Start | GRPO | PTST | Avg. Len. | Avg. Acc. |
|---|---|---|---|---|---|
| Vision-R1-Zero | | ✓ | | 1285 | 50.7 |
| Vision-R1-CI | ✓ | | | 3566 | 44.5 |
| Vision-R1-Long | ✓ | ✓ | | 3107 | 47.7 |
| Vision-R1 (Ours) | ✓ | ✓ | ✓ | 2057 | 55.4 |

Moreover, on the complex math reasoning sub-tasks of MathVista, *i.e.*, GEO, ALG, and GPS, Vision-R1-7B achieves scores of 80.3%, 79.0%, and 83.2%, respectively, exceeding the base model Qwen-2.5-VL-7B by an average accuracy improvement of over 10%. These results highlight the strong reasoning capability Vision-R1-7B gains through "human-like" complex thinking processes. Furthermore, on the more challenging MathVerse and MM-Math benchmarks, Vision-R1-7B ranks Top-1 and Top-2, respectively, with the latter being second only to Qwen-2.5-VL-72B. This demonstrates Vision-R1-7B's effectiveness in solving complex math problems.

### 4.2 ABLATION STUDY

**Cold-start Dataset Quality.** We conduct a quality analysis of our proposed Vision-R1-cold dataset. The primary objective of constructing Vision-R1-cold is to supplement the existing multimodal CoT datasets, which lack complex cognitive processes, and to leverage DeepSeek-R1's high-quality CoT process as cold-start data. To evaluate this, we present a comparative analysis in Tab. 4, where we statistically examine the presence of questioning, reflection, and inspection within the CoT processes of Mulberry, LLaVA-CoT, and Vision-R1-cold datasets. The results indicate that Vision-R1-cold contains a significantly higher proportion of human-like cognitive processes compared to previous multimodal CoT datasets. This complex CoT structure facilitates the base MLLM in learning reasoning mechanisms, providing a high-quality cold-start initialization for RL training.

Additionally, we compare Vision-R1-cold with previous multimodal CoT datasets by training on Llama-3.2-11B-V in Tab. 2. After SFT, our Vision-R1-LlamaV-CI-11B model achieves SoTA

Table 4: Comparison of the occurrence frequency of self-reflective indicators between llava-cot, mulberry and our Vision-R1-cold dataset. The higher frequency of these reflective markers in our dataset demonstrates its distinctive self-reflection and self-correction characteristic.

| Word | llava-cot (100K) | Mulberry (260K) | Vision-R1-cold (200K) |
|------|------|------|------|
| "Wait" | 2,300 | 1,122 | 585,719 |
| "Hmm" | 1 | 0 | 75,853 |
| "Mistake" | 183 | 8,784 | 26,697 |
| "Alternatively" | 251 | 68 | 188,187 |
| "Check" | 8,332 | 26,421 | 100,148 |

Table 5: Effect of PTST. "4K×16" denotes a PTST setup where the Stage limits the response length to 4K with 16 samples. "Avg." denotes the average performance (MathVista, MathVerse, MM-Math).

| Method | Stage 1 | Stage 2 | Stage 3 | MathVisita | MathVerse | MM-Math | Avg. |
|--------|---------|---------|---------|-----------|-----------|---------|------|
| Baseline | – | – | – | 68.1 | 46.7 | 34.1 | 49.6 |
| – | 4K×16 | 4K×16 | – | 72.6 | 51.4 | 39.0 | 54.3 |
| – | 4K×16 | 8K×16 | – | 72.9 | 53.5 | 39.5 | 55.3 |
| – | 4K×16 | 6K×12 | 8K×8 | 73.0 | 52.6 | 39.8 | 55.1 |
| Vision-R1-Long | 16K×4 | 16K×4 | – | 70.3 | 36.8 | 36.1 | 47.7 |
| – | 16K×16 | 16K×16 | – | 71.1 | 36.5 | 36.1 | 47.9 |
| Vision-R1 (Ours) | 4K×16 | 8K×8 | – | 73.5 | 52.4 | 40.2 | 55.4 |

performance across all general and math reasoning benchmarks, outperforming both LLaVA-CoT-11B and Mulberry-Llama-11B, directly confirming the superior quality of Vision-R1-cold dataset.

**Effect of Main Strategy.** As shown in Tab. 3, we compare the performance of various RL training strategies. The results indicate that Vision-R1-Zero, which applies RL training directly without cold-start initialization, struggles to generate sufficiently long and complex CoT reasoning, thereby limiting its ability to handle intricate reasoning tasks. In contrast, Vision-R1-CI, after cold-start initialization, tends to generate excessively long CoTs. However, the presence of numerous incorrect reasoning steps leads to lower overall performance. Furthermore, Vision-R1-Long via applying RL training directly to Vision-R1-CI results in optimization difficulty, making it hard to achieve significant performance improvements. In comparison, our proposed Vision-R1 demonstrates a substantial advantage in reasoning performance, effectively balancing CoT complexity and accuracy.

**Effect of PTST.** In Tab. 5, our two-stage setup (4K×16→8K×8) achieves the best average (55.4%), outperforming training with a fixed short length (4K×16→4K×16) by +1.1 Avg. (54.3→55.4), indicating that progressively relaxing the length constraint in Stage 2 yields consistent gains once correct thinking is established in Stage 1. In contrast, training with fixed 16K (16K×4 or 16K×16) severely underperforms (47.7%/47.9%), showing that early length constraints effectively mitigate overthinking. Further increasing Stage 2 sampling to 16 (4K×16→8K×16, 55.3%) or inserting an extra stage (4K×16→6K×12→8K×8, 55.1%) brings no meaningful improvement, suggesting PTST is robust and a simple two-stage schedule suffices under matched training time (sampling×length kept constant per stage).

**Effect of Cold Start.** Tab. 6 demonstrates that PTST alone offers limited benefit without cold start (Zero+PTST: 51.8% vs Vision-R1-Zero: 50.7%), and SFT without CoT before RL is detrimental (Zero+SFT+PTST: 39.8%). In contrast, cold-starting on Vision-R1-cold followed by PTST yields the best average (55.4%), with a substantial gain on MM-Math (40.2% vs 33.1%/28.8%). These results support that complex CoT priors acquired in cold start are essential for RL to learn correct reasoning patterns. Furthermore, PTST then suppresses early overthinking (short 4K) and safely extends reasoning length (8K), leading to consistent improvements.

**Visualization.** As shown in Fig. 4, our proposed Vision-R1-7B is capable of generating complex reasoning processes and exhibits the emergence of the so-called "Aha moment" (DeepSeek-AI, 2025), *i.e.*, a phenomenon analogous to human cognitive processes involving questioning and reflection. This sophisticated reasoning capability significantly enhances the model's inference performance, leading to substantial improvements in solving complex reasoning tasks.

Table 6: Effect of Cold Start. "Zero+PTST" denotes we keep the Vision-R1-Zero settings while adopting PTST strategy. "Zero+SFT+PTST " involves training the base model in the Vision-R1-cold dataset *without CoT annotations* before RL training, followed by RL training with the same "Zero+PTST" setup. "Avg." denotes the average performance (MathVista, MathVerse, MM-Math).

| Method | MathVisita | MathVerse | MM-Math | Avg. |
|---|---|---|---|---|
| Vision-R1-Zero | 70.6 | 52.6 | 28.8 | 50.7 |
| Zero+PTST | 71.3 | 50.9 | 33.1 | 51.8 |
| Zero+SFT+PTST | 68.7 | 32.0 | 18.9 | 39.8 |
| Vision-R1 (Ours) | 73.5 | 52.4 | 40.2 | 55.4 |

## 5 CONCLUSION

We explore how to use RL training to incentivize reasoning capability in MLLMs. Moreover, we proposed Vision-R1 and achieved the strong math reasoning ability, achieving comparable performance to SoTA MLLMs.

## ACKNOWLEDGEMENTS

This work is supported by the National Natural Science Foundation of China (NO. 62572193), the Open Research Fund of the Key Laboratory of Advanced Theory and Application in Statistics and Data Science, Ministry of Education, and the Fundamental Research Funds for the Central Universities.

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

# A    EXPERIMENT SETTINGS

**Dataset and Benchmarks.** To obtain the cold-start dataset, we use the multimodal visual question answering (VQA) datasets, LLaVA-CoT dataset (100K) (Xu et al., 2024) and Mulberry dataset (260K) (Yao et al., 2024), to conduct Vision-R1-cold (200K). During GRPO process, we mix the math datasets of We-Math (Qiao et al., 2024), MathVision (Wang et al., 2025), Polymath (Gupta et al., 2024), SceMQA (Liang et al., 2024), Geometry3K (Lu et al., 2021a) as our RL training data. Total amount of data is around 10K. For scaling up the RL dataset for Vision-R1-32B and Vision-R1-72B, we render text-based AIME data before 2024 as images, extract a subset from MAmmoTH-VL (Guo et al., 2024) and MMIQ (Cai et al., 2025), obtaining a total of ∼20K additional data.

For evaluating the reasoning capability of our Vision-R1, we choose three widely used multimodal math benchmarks: MM-Math (Sun et al., 2024), MathVista (Lu et al., 2023a), MathVerse (Zhang et al., 2024a). These benchmarks covers various mathematical fields which can provide a thorough evaluation of MLLMs' math reasoning ability. Besides, we select four general multimodal benchmarks to demenstrate the general ability of our model: MMStar (Chen et al., 2024b), ChartQA (Masry et al., 2022), MME (Fu et al., 2023) and HallBench (Guan et al., 2024). Those general benchmarks are used to evaluate the data quality of our proposed Vision-R1-cold dataset.

**Implementation Details.** For the Vision-R1-cold dataset preparation, we deploy the open-source MLLM Qwen2.5-VL-72B (Bai et al., 2025) and the reasoning LLM DeepSeek-R1 (DeepSeek-AI, 2025). We then process the VQA datasets using Qwen-2.5-VL-72B and DeepSeek-R1.

For the cold-start initialization of Vision-R1, we adopt Qwen2.5-VL (Bai et al., 2025) as the base model and train it via supervised fine-tuning (SFT) for 2 epochs using Llama-Factory framework (Zheng et al., 2024). After cold-start initialization, we obtain the post-cold-start model, Vision-R1-CI, which is subsequently trained on the collected math dataset using GRPO in Verl training framework (Sheng et al., 2024; Zheng et al., 2025), while following the two-stage PTST approach (note that we do not use the third stage checkpoints), *i.e.*, $S$ was set to 2. The all models in the paper are summarised as follows:

- **Vision-R1-Zero**: This represents the baseline approach where reinforcement learning (RL) is applied directly to the base MLLM without cold-start initialization. We adopt the limit of 4K generation token length with 16 samples to train the base model via RL training. The system prompt and the training method are described in Sec. 3.1. The full training step is set to 300.

- **Vision-R1-CI**: The base MLLM is cold-start initialized using the Vision-R1-cold dataset, resulting in this model.

- **Vision-R1-Long**: This variant is trained with a maximum generation length of 16K tokens, where four samples per input are generated from the cold-started Vision-R1-CI, followed by 300 training steps.

- **Vision-R1**: This model follows the Progressive Thinking Suppression Training (PTST) strategy, where a two-stage RL training process is applied:
  - **Stage 1**: The model is trained from Vision-R1-CI for 100 steps with an 4K token generation limit, sampling 16 samples per input.
  - **Stage 2**: Training continues for another 100 steps with a maximum generation length of 8K tokens, sampling 8 samples per input.

  The final model checkpoint at the end of Stage 2 is the final Vision-R1 model, as this stage achieves an optimal balance between reasoning length and overall performance. Note that for **Vision-R1-32B** and **Vision-R1-72B**, we use additional data to continue training them in the RL stage under the same PTST Stage 2 settings.

Additionally, Vision-R1 can be further extended to a third training stage, following the same parameter settings as Vision-R1-Long and continuing for an additional 100 training steps. However, as shown in Fig. 1, further training does not yield significant performance improvements but does generate more complex reasoning processes.

For the ablation study experiments in Sec 4.2, we choose Vision-R1-7B as the default to evaluate. In Tab. 5 and Tab. 6, we chose the optimal checkpoint for reporting performance by evaluating every

Simple Description

The image is a bar chart displaying the infant mortality rate in Kiribati from 2009 to 2019. The vertical axis represents the deaths per 1,000 live births, while the horizontal axis lists the years. Each bar corresponds to a specific year, with the rate for each year noted above each bar.

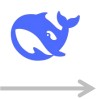

DeepSeek-R1

I need to check......Wait, but **the original data isn't here**......Wait, maybe **there's a mistake**......Hmm, **this is a problem**......Alternatively, **maybe** the answer is "34.2" as per real data, but **I can't confirm**......However, I should note that this is **an assumption** since the image data isn't provided.

---

Detailed Description

The image is a bar chart displaying the infant mortality rate in Kiribati from 2009 to 2019...... **The rates for each year are as follows: 49.9 in 2009......40.1 in 2019.** The bar for 2019 clearly indicates the infant mortality rate as 40.1 deaths per 1,000 live births.

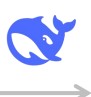

DeepSeek-R1

Okay, let's see. The question......First, I'll recall the data points given......I need to confirm......Let me check again......So, **the answer should be straightforward**......which is **clearly stated as** 40.1.......I should make sure......Therefore, the answer is 40.1.

Figure 5: Comparison between the CoT processes generated by descriptions with and without "Pseudo-CoT". Simple descriptions generated without the "Pseudo-CoT" input lack sufficient visual information, leading to confusion and hallucination in the reasoning process of DeepSeek-R1. In contrast, detailed descriptions enhanced through our Modality Bridging with "Pseudo-CoT" integrate high-quality visual information into textual descriptions, which facilitates accurate reasoning and enables R1 to generate correct answers.

fifth step in the final 50 steps of the RL training process, thereby avoiding the statistical bias caused by training instability.

To assess the quality of the Vision-R1-cold dataset, we apply the same training hyper-parameters of the cold-start initialization to Llama-3.2-V-Instruct (Dubey et al., 2024) for SFT, resulting in another post-cold-start MLLM, **Vision-R1-LlamaV-CI**, which can also be used for subsequent RL training.

## B    CoT Comparison for Data Construction Strategy

We compare the CoT example generated by descriptions with and without "Pseudo-Co" in Fig. 5. When we directly use the naive image description as the input for DeepSeek-R1, the strong reasoning LLM cannot generate the correct trajectories for a massive amount of VQA data. This is because the MLLM does not know which image information is important for reasoning, and the generated image description lacks key converted image information, resulting in unusual CoT data.

## C    Additional Dataset Illustrations

We provide some more examples of our Vision-R1-cold dataset in Fig. 6. Fig. 7 showcases multiple response instances generated by our Vision-R1 model, including comprehensive reasoning processes and corresponding final answers.

## D    Data Sources

Our proposed Vision-R1-cold dataset consists of a wide range of multimodal data, including the following categories:

- Mathematical Data: GLLaVA (Gao et al., 2023), GEOS (Seo et al., 2015), UniGeo (Chen et al., 2022), GeoQA Plus (Chen et al., 2021), Geo3K (Lu et al., 2021a), MathVision (Wang

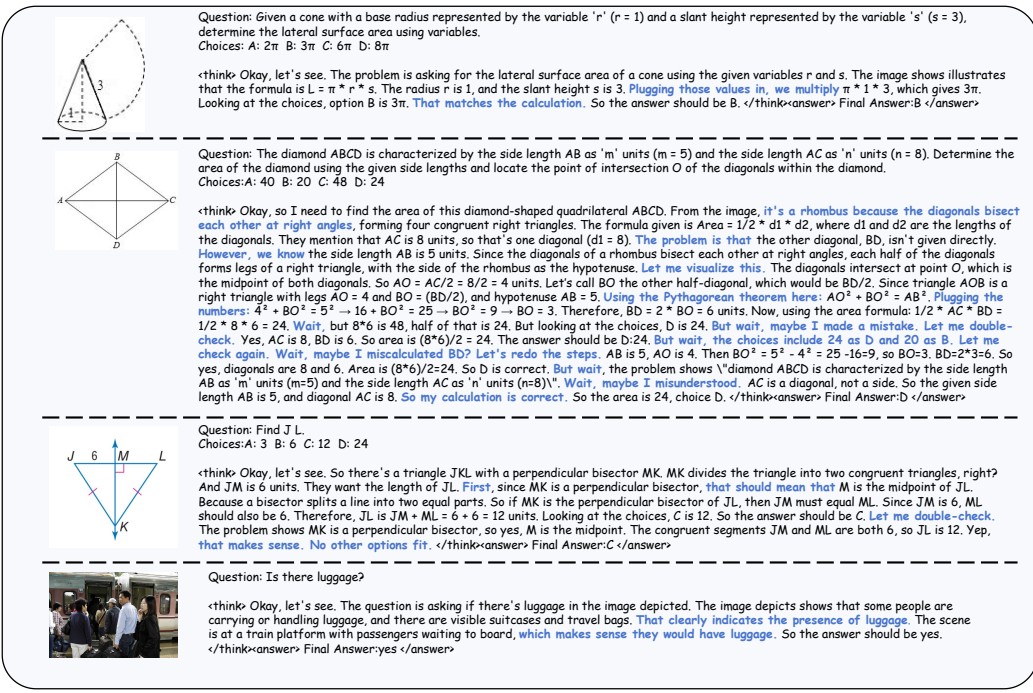

Figure 6: Examples of our Vision-R1-cold data. It comprises abundant information obtained through our Modality Bridging method.

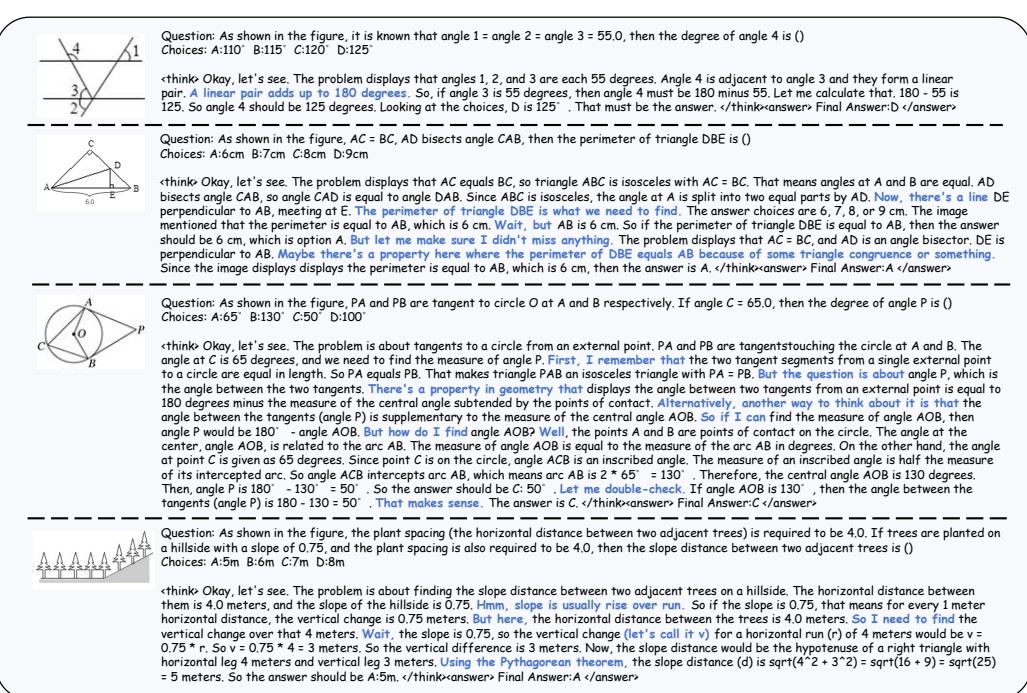

Figure 7: More output examples of our Vision-R1-7B on MathVerse benchmark.

et al., 2025), GeoMverse (Kazemi et al., 2023), MathV360K (Shi et al., 2024), IconQA (Lu et al., 2021b), TabMWP (Lu et al., 2023b), CLEVR (Johnson et al., 2017), CLEVR-Math (Lindström & Abraham, 2022), and Super-CLEVR (Li et al., 2023).

- General QA Data: ShareGPT4V (Chen et al., 2024a), PISC (Li et al., 2017) VQA-AS (Antol et al., 2015), A-OKVQA (Schwenk et al., 2022), TextVQA (Singh et al., 2019), Vizwiz (Gurari et al., 2018), and VQA2.0 (Goyal et al., 2017)

- Science and Medical Data: From GeoQA+ (Cao & Xiao, 2022), CLEVR-Math (Lindström & Abraham, 2022), TQA (Kembhavi et al., 2017), AI2D (Kembhavi et al., 2016), ScienceQA (Lu et al., 2022), VQA-RAD (Lau et al., 2018), and PMC-VQA (Zhang et al., 2023)

- Figure Understanding Data: From DVQA (Kafle et al., 2018), DocVQA (Mathew et al., 2021), FigureQA (Kahou et al., 2017), PlotQA (Methani et al., 2020), ChartQA (Masry et al., 2022), InfoVQA (Mathew et al., 2022), MultiHiertt (Zhao et al., 2022), and LRV-Chart (Liu et al., 2023).

## E    LLM USAGE

In this paper, we have used an LLM to refine some sentences and improve the grammar, making the paper more academic.

