# OpenReview forum: "Vision-R1: Incentivizing Reasoning Capability in Multimodal Large Language Models"
_ICLR.cc/2026/Conference — ICLR 2026 Poster_

### Official Review · Reviewer_Lj3M · 2025-10-27

**Soundness:** 2
**Presentation:** 2
**Contribution:** 2
**Rating:** 4
**Confidence:** 4

**Summary:**

The paper introduces ​​Vision-R1​​, a method to enhance reasoning in ​​Multimodal Large Language Models (MLLMs)​​ by combining ​​cold-start initialization​​ with ​​Reinforcement Learning (RL)​​.

This paper proposes a pipeline to build high-quality multimodal CoT datasets without human annotation, using MLLMs and DeepSeek-R1 and uses this pipeline to build Vision-R1-cold (200K samples)​​.

​​This paper proposed Progressive Thinking Suppression Training (PTST)​​ to mitigate ​​overthinking​​ in RL training by gradually increasing reasoning complexity and analyzes differences between direct RL training and the combined approach of coldstart initialization and RL training.

**Strengths:**

​1. ​High-quality dataset construction:​​ The ​​Vision-R1-cold dataset​​ leverages ​​modality bridging​​ to generate ​​human-like CoT reasoning​​ without manual annotation, addressing a critical bottleneck.


2. ​​Strong performance:​​ Vision-R1-7B ​​matches or surpasses larger models​​ (e.g., Qwen2.5-VL-72B) on ​​MathVista, MathVerse, and MM-Math​​, demonstrating efficiency and scalability.

**Weaknesses:**

1. The novelty is limited. The article is more engineering-oriented, and the overall innovation of the work is incrimental.

2. The argument for the PTST method's effectiveness in mitigating the Overthinking problem is insufficient. The paper uses the results of Vision-R1-Long to reveal the Overthinking problem and compares it with Vision-R1 (w/ PTST) to demonstrate that PTST alleviates the issue. However, the length limit for Vision-R1-Long is 16K, while Vision-R1 (w/ PTST) has limits of 4K and 8K in its two stages. If the length limit for Vision-R1-Long were 8K (or 4K), would the Overthinking problem still exist? How would the comparison with Vision-R1 (w/ PTST) look then?

3. The high-quality dataset in the Cold Start phase is one of the paper's key contributions, yet the necessity of Cold Start is not sufficiently justified. The proposed method consists of two phases (see Table 4): Cold Start and GRPO (w/ PTST). The paper does not seem to conduct ablation experiments on Cold Start. Specifically, in Table 4, Vision-R1-Zero performs better than Vision-R1-Long, which undergoes Cold Start in addition to Vision-R1-Zero. Does this imply that skipping Cold Start is better? I believe that conducting an experiment with GRPO (w/ PTST) (without Cold Start) and comparing it with the existing Vision-R1 would validate the necessity of Cold Start.

4. The paper claims that one of its contributions is analyzing the difference between direct RL training and the combined approach of Cold Start initialization and RL training. The former corresponds to the Vision-R1-Zero model in the paper, but the training configuration (methods, parameters, etc.) for Vision-R1-Zero is not detailed. Additionally, the paper compares Vision-R1-Zero with Vision-R1-CI, but their training data differs. I think a more meaningful comparison would involve training Vision-R1-Zero with the same data as Vision-R1-CI before contrasting them.

5. The ablation experiments for PTST are insufficient. PTST primarily involves two settings: gradually increasing the sequence length limit $L_s$ and gradually decreasing the sampling count $G_s$. However, the paper does not conduct ablation experiments for either setting. For example, why should the sampling count $G_s$ gradually decrease? What would happen if it remained constant or gradually increased?

**Questions:**

Seeing weaknesses

---

> ### Author Response · Authors · 2025-11-12
>
> Dear Reviewer Lj3M,
>
> Thank you very much for your efforts in reviewing our paper and for providing valuable feedback.
>
> We notice that your review appears to be **consistent with the one from the previous conference cycle**. However, we would like to emphasize that our manuscript has undergone substantial revisions and enhancements, including comprehensive ablation studies, detailed experimental settings, and scaling up of both data and models.
>
> Some of the concerns you raised may have already been addressed in the current version or may not align with the revised manuscript. Specifically:
>
> 1. The "Table 4" mentioned in your review likely corresponds to Table 3 in the current version.
> 2. Regarding W2 concerning the "Effect of PTST": relevant content can be found in Table 5 and its corresponding section.
> 3. Regarding W3 concerning the "Effect of Cold Start": relevant content can be found in Table 6 and its corresponding section.
> 4. Regarding W4 concerning "Training setting" and "Different data": these are addressed in Appendix A and Table 6, respectively.
> 5. Regarding W5 concerning "More detailed ablation study of PTST setting": this can also be found in Table 5.
>
> We are currently preparing a comprehensive, point-by-point response to address all your concerns. However, before proceeding, we would greatly appreciate it if you could re-evaluate our submission based on the current ICLR version and inform us of any additional concerns or questions.
>
> We look forward to your feedback.
>
> Sincerely Yours,
>
> The Authors

---

> > ### Comment · Reviewer_Lj3M · 2025-11-13
> >
> > Thank you for your message and for the clarifications regarding the revisions made in the current version of your manuscript.
> >
> > I appreciate your efforts in substantially updating the submission with additional experiments, ablations, and detailed settings. I also acknowledge that some of my earlier comments may have overlapped with concerns raised in the previous review cycle.
> >
> > I realize that I overlooked some of the detailed revisions in the current version. I will carefully re-examine the paper. In the meantime, I would appreciate it if you could provide detailed responses to the points that have not yet been addressed, particularly those related to novelty.
> >
> > Thank you again for the clarifications, and I appreciate the work you have put into improving the paper. I am willing to raise my score to 6, and if you can further convince me regarding the novelty aspect, I would be open to increasing it again.

---

> > > ### Author Response · Authors · 2025-11-28
> > >
> > > Dear Reviewer Lj3M,
> > >
> > > Thank you again for your invaluable time and effort in reviewing our paper. We sincerely appreciate your acknowledgment that some of your earlier comments may have overlapped with concerns raised in the previous review cycle. We are grateful that you took the time to re-review our ICLR submission and raised the score to 6. We deeply appreciate your recognition of our work!
> > >
> > > We have also prepared a detailed rebuttal to address any remaining concerns you may have. We hope this will help clarify any outstanding issues.
> > >
> > > Sincerely yours,
> > >
> > > The Authors

---

> ### Author Response · Authors · 2025-11-27
> **Rebuttal (1/2)**
>
> > W1: Novelty and Contribution
>
> We emphasize that our contribution is exploring **how to effectively adapt existing techniques to MLLMs and address domain-specific challenges**. We provide substantial insights, including:
>
> - DeepSeek-R1 achieved remarkable success in incentivizing reasoning capability in LLMs, can this success transfer to the MLLM domain?
> - Directly applying RL to MLLMs does not replicate DeepSeek-R1's success. Given the insufficient base model capabilities and limited data in the multimodal domain, what prevents effective reasoning emergence? Is it the lack of complex "human-like" CoT?
> - If we need to inject complex reasoning generation capabilities involving reflection and questioning into MLLMs, how should we construct effective cold-start data?
> - After cold-start, MLLMs exhibit complex reasoning but suffer from performance degradation and overthinking phenomena that hinder subsequent optimization, how can we effectively elicit reasoning capability during the RL phase?
>
> **These questions for reasoning MLLMs were unexplored and unanswered prior to our work**, and we provide effective solutions. This parallels the contribution of LLaVA [1]: when instruction tuning was successfully applied to LLMs, the MLLM domain still faced unique challenges. Exploring and solving domain-specific problems constitutes significant contribution. **Effectively leveraging existing techniques does not imply lack of novelty**, we have made substantial contributions to multimodal reasoning and established a standard practice for this domain. Therefore, we firmly believe our work significantly advances the field and represents **far more than an "incremental contribution"**.
>
> We summarize our key contributions as follows:
>
> - **First work to explore how to effectively incentivize complex "human-like" reasoning capability in MLLMs**, identifying and addressing challenges in building complex reasoning MLLMs, thereby establishing a standard practice for the reasoning MLLM domain.
> - **A novel pipeline for constructing high-quality multimodal CoT datasets without human annotation**, producing the 200K Vision-R1-cold dataset with complex CoT for model cold-start. We demonstrate its effectiveness across different model sizes (7B/32B/72B) and data scales, validating it as a high-quality, scalable data synthesis methodology and dataset.
> - **Identification of the overthinking problem in cold-started MLLMs** and proposal of a simple yet effective RL training strategy with progressive thinking suppression that gradually relaxes reasoning length constraints. This approach proves highly effective for scaling up scenarios.
>
> As discussed in before, our method establishes a **standard practice for the domain**. We argue that exploring a simple yet effective solution to a novel problem and validating its effectiveness constitutes a valuable community contribution. Solutions to current challenges need not be complex. When an intuitive approach suffices, subsequent work can build upon our insights to propose more sophisticated methods for further improvements. Therefore, **we believe this does not diminish our contribution**.

---

> ### Author Response · Authors · 2025-11-27
> **Rebuttal (2/2)**
>
> > **W2: Effectiveness of PTST**
>
> We thank the reviewer for this suggestion.
>
> As stated in L316-318, the key to solving the overthinking problem is **"guiding the model to learn correct thinking in the early stages for the reasoning performance of MLLMs"**. The overthinking phenomenon is observed when cold-started models produce correct responses predominantly at shorter lengths, while excessively long reasoning processes tend to be erroneous and redundant.
>
> Without PTST constraining response length in early RL training to guide correct reasoning (e.g., Vision-R1-Long), models continue to generate excessively long redundant reasoning (as observed in cold-started models), thereby **increasing optimization difficulty and hindering performance**. Furthermore, Table 5 demonstrates that aligning Vision-R1-Long's sample count with Vision-R1 Stage 1 (using 16 samples) shows that even with increased samples, failing to suppress response length early on prevents effective optimization of cold-started models.
>
> *We provide additional discussion and details in our **response to W4 of reviewer XezA***.
>
>
> > **W3: Effectiveness of Cold-Start**
>
> First, we emphasize that **Vision-R1-Long's lower performance compared to Vision-R1-Zero does not imply skipping cold-start is better**. Rather, it demonstrates that **applying PTST during RL for correct learning is an indispensable step**.
>
> As shown in Table 3, the cold-started Vision-R1-CI achieves an average score of 44.5 on MathVista, MathVerse, and MM-Math, lower than the base model's 49.6 before cold-start. Building on this foundation, when we apply RL to Vision-R1-CI without addressing overthinking through early length constraints, we fail to effectively elicit reasoning capability (Vision-R1-Long achieves 47.7 average, showing some improvement over Vision-R1-CI but remaining modest). **Only when properly combined with PTST does the model demonstrate the significant advantages of complex reasoning patterns learned through cold-start compared to direct RL**.
>
> Moreover, as shown in Table 6, we apply PTST to Vision-R1-Zero, but it still cannot achieve the reasoning capability gains obtained through cold-start.
>
> *We provide additional discussion and details in our **response to W2 of reviewer XezA***.
>
>
> > **W4: Detailed Training Settings and Training Data Ablation**
>
> Detailed training settings can be found in Appendix A.
>
> Additionally, Table 6 presents data ablation results. We first perform SFT training on the base MLLM using Vision-R1-cold dataset without CoT annotations, then conduct RL training. As shown, when controlling for identical VQA data exposure, training without complex CoT significantly degrades model performance. This phenomenon stems from the limited quality and scale of the VQA data, demonstrating that **Vision-R1's substantial performance gains originate from learned complex reasoning processes rather than additional data**.
>
>
> > **W5: Additional Ablations on PTST**
>
> We thank the reviewer for this valuable feedback.
>
> Regarding our strategy of gradually increasing sequence length limits while gradually decreasing sampling counts: PTST employs different response lengths and sample numbers across stages while maintaining a consistent maximum token budget (e.g., maximum tokens per prompt rollout is constrained to 64K tokens in both Stage 1 "4K×16" and Stage 2 "8K×8"). This design ensures that **average rollout overhead remains approximately constant across PTST stages**.
>
> Table 5 presents ablation experiments on response length and sample count. When maintaining Stage 2's sampling count identical to Stage 1 (e.g., "8K×16" in Stage 2), the model achieves performance comparable to original PTST **without significant improvement**. Considering the additional computational overhead from increased sample counts, **gradually decreasing sample counts achieves a favorable balance between performance and computational cost**.
>
> > **Reference**
>
> [1] Visual instruction tuning. NeurIPS 2023.

---

### Official Review · Reviewer_XezA · 2025-10-27

**Soundness:** 3
**Presentation:** 4
**Contribution:** 3
**Rating:** 4
**Confidence:** 5

**Summary:**

This paper introduces a training paradigm aimed at enhancing the reasoning capabilities of Multimodal Large Language Models. The authors first construct a 200K multimodal CoT dataset, named Vision-R1-cold, without requiring manual human annotation. They then propose a Progressive Thinking Suppression Training strategy, which progressively relaxes the upper limit on reasoning length during RL training. The method demonstrates performance improvements on several multimodal math reasoning benchmarks, including MathVista and MathVerse.

**Strengths:**

1. The paper's "Modality Bridging" method is a strength. It uses an existing MLLM to transform visual information into detailed textual descriptions. This enables a powerful text-only model (DeepSeek-R1) to generate high-quality, human-like CoT for multimodal data.
2. The models achieve competitive performance on benchmarks like MathVista.
3. The paper is clearly written, and the figures generally present the concepts effectively.

**Weaknesses:**

1. The Vision-R1-cold dataset is constructed by filtering samples from various sources, including LLaVA-CoT and Mulberry. The paper does not provide a clear analysis to rule out potential data leakage or overlap between these source datasets and the evaluation benchmarks (e.g., MathVista, MathVerse). This lack of a contamination check casts some doubt on the true generalization performance.

2. The definition of "Pseudo-CoT" is perplexing. The authors initially introduce it as a negative, "formatted" reasoning style lacking true cognition. However, they then use this "Pseudo-CoT" as a critical component in their own data pipeline (Figure 2) to generate descriptions. This leads to a critical missing ablation study: What is the performance if one simply uses the "Pseudo-CoT" data directly for cold-start initialization? The paper does not compare SFT with the "Pseudo-CoT" data. This experiment is essential to prove that the complex, DeepSeek-R1-generated CoT is superior and that the complex data pipeline is necessary. Finally, this paper has a coupling problem: The performance gain from the cold-start data might come from two sources: (1) the data source selection process or (2) the complex CoT style. The paper fails to decouple these effects.

3. The proposed Progressive Thinking Suppression Training is presented as a new strategy, but its technical contribution is limited.

4. The paper successfully identifies and names the "Overthinking Optimization Problem", but the analysis of its root cause is shallow. The authors state that after SFT on complex CoT, correct answers are concentrated in shorter sequences. Why does this happen? Is the model merely imitating the complex style of CoT without grasping the reasoning? Or is this problem caused by the data itself or the SFT training dynamic? Crucially, would this "overthinking" problem still occur if the (simpler) "Pseudo-CoT" data were used for cold-start instead? The paper fails to investigate these critical questions.

5. The "Modality Bridging" pipeline (MLLM -> Caption -> DeepSeek-R1) has a potential fundamental flaw. An MLLM's ability to "caption" an image with perfect, fine-grained detail is limited. The authors' method attempts to guide this by feeding the MLLM a "Pseudo-CoT" that already contains reasoning steps. This process, which is inherently "answer-directed," could introduce a bias or lead to a feedback loop of hallucinations, where the MLLM generates a caption that justifies the answer rather than describing the image objectively. The paper does not analyze or mitigate this risk.

**Questions:**

Same as the content in Weaknesses.

---

> ### Author Response · Authors · 2025-11-27
> **Rebuttal (1/4)**
>
> > **W1: Potential Data Contamination**
>
> We thank the reviewer for this important concern.
>
> We address this with multiple lines of evidence demonstrating that our performance gains stem from improved generalization capability rather than data leakage in the cold-start dataset:
>
> **Evidence 1: Cold-start alone does not improve performance.**
> As shown in Table 3, Vision-R1-CI, after cold-start with the Vision-R1-cold dataset, does not improve but rather degrades baseline performance (49.6→44.5). The final performance gains are attributed to learning complex reasoning patterns.
>
> **Evidence 2: Identical data sources with different CoT styles yield different outcomes.**
> Our cold-start VQA data originates from LLaVA-CoT-100K [1] and Mulberry-260K [2]. As shown in Table 2, when cold-starting Llama-3.2-11B-V, Vision-R1-cold dataset (using the same source data but with synthesized complex CoT), demonstrates **significant performance improvements** over models fine-tuned with LLaVA-CoT-100K and Mulberry-260K. This substantiates that the primary gains arise from "human-like" reasoning processes.
>
> **Evidence 3: Quantitative similarity analysis reveals minimal overlap.**
> We examined potential leakage in Vision-R1-cold dataset by extracting image features using Qwen-2.5-VL (average pooling of last-layer features after pre-filling) and computing cosine similarity with MathVista and MathVerse images (identical data should yield 1.0 similarity). The following table shows the proportion of samples exceeding certain similarity thresholds:
>
> |           | Similarity > 95% | Similarity > 99% | Similarity = 100% |
> | --------- | ---------------- | ---------------- | ----------------- |
> | MathVista | 0.23%            | 0.06%            | 0%                |
> | MathVerse | 0%               | 0%               | 0%                |
>
> Manual inspection of samples with >95% and >99% similarity reveals they typically share background context but with modified critical information (e.g., altered edge lengths in geometry problems). This indicates some similar problems exist between LLaVA-CoT-100K/Mulberry-260K and MathVista, but **we found no perfectly identical problems**. Moreover, MathVerse shows no high-similarity samples in Vision-R1-cold, yet still achieves significant performance improvements (Table 1).
>
> **Evidence 4: Consistent improvements across diverse benchmarks.**
> Data contamination typically causes pronounced improvements on specific benchmarks. However, our experiments demonstrate **consistent significant improvements across 4+ mathematical benchmarks**. *As shown in our response to **W1 of reviewer i3eD***, we also observe generalization improvements on multiple non-mathematical benchmarks.
>
> Based on these findings, we conclude that **Vision-R1's performance primarily stems from enhanced generalization capability rather than data leakage**.

---

> > ### Author Response · Authors · 2025-11-27
> > **Rebuttal (2/4)**
> >
> > > **W2: Ablation on "Pseudo-CoT" and Performance Gains from Cold-Start Data**
> >
> > First, we emphasize that Table 2 in our paper presents a direct comparison on Llama-3.2-11B-V, contrasting models cold-started with Vision-R1-cold dataset versus those fine-tuned with "Pseudo-CoT" methods (LLaVA-CoT-100K and Mulberry-260K). The results demonstrate that simply replacing the CoT with "human-like" complex reasoning processes, while using VQA data from the same sources, yields significant performance improvements, **indicating that gains originate from the complex CoT style**.
> >
> > Furthermore, we conduct additional experiments using identical Vision-R1 settings to train models cold-started with "Pseudo-CoT" data, further elucidating the source of performance gains.
> >
> > We cold-start the MLLM with Mulberry-260K using hyperparameters identical to Vision-R1-CI's cold-start process. Subsequently, we apply RL to the cold-started model. To accommodate Mulberry-260K's data format (which specifies output format in questions), we employ the following system prompt:
> >
> > ```
> > Generate an image description based on the question.\nThen, provide a rationale to analyze the question.\nNext, generate a step-by-step reasoning process to solve the problem. Ensure the steps are logical and concise.\nFinally, provide a concise summary of the final answer in the following format: 'The final answer is: xxx.\n\nFormat your response with the following sections, separated by ###:\n### Image Description:\n### Rationales:\n### Let's think step by step.\n### Step 1:\n### Step 2:\n...\n### The final answer is: \n
> > ```
> >
> > During RL training, we maintain identical settings as Vision-R1, including the PTST strategy and all hyperparameters, while using the hard formatting result reward function (reward=1 when both formatting requirements and answer correctness are satisfied). Results are shown below, where "Vision-R1-CI w/ mulberry-260k" and "Vision-R1 w/ mulberry-260k" denote the cold-started and RL-trained models, respectively:
> >
> > |                               | MathVista | MathVerse | MM-Math  | Avg.            |
> > | ----------------------------- | --------- | --------- | -------- | --------------- |
> > | Qwen-2.5-VL-7B                | 68.1      | 46.7      | 34.1     | 49.6            |
> > | Vision-R1-CI w/ mulberry-260k | 64.4      | 48.5      | 20.1     | 44.3 (-5.3)     |
> > | Vision-R1-CI                  | 68.8      | 39.2      | 25.6     | 44.5 (-5.1)     |
> > | Vision-R1 w/ mulberry-260k    | 68.4      | 51.6      | 25.6     | 48.5 (-1.1)     |
> > | **Vision-R1 (ours)**          | **73.5**  | **52.4**  | **40.2** | **55.4 (+5.8)** |
> >
> > As demonstrated, when using "Pseudo-CoT" for cold-start followed by RL training, **the reasoning processes learned from step-level "Pseudo-CoT" fail to achieve effective performance improvements**. We also present training and validation set rewards during RL (using identical settings as Vision-R1):
> >
> > |                            | Training Set Reward (Acc.) | Val Set Reward (Acc.) |
> > | -------------------------- | -------------------------- | --------------------- |
> > | Vision-R1 w/ mulberry-260k | 0.64                       | 0.48                  |
> > | Vision-R1 (ours)           | 0.61                       | 0.44                  |
> >
> > Notably, Vision-R1 w/ mulberry-260k achieves higher accuracy rewards on both sets compared to Vision-R1. However, **due to its inability to generate reasoning processes with complex reflection, it fails to generalize to performance improvements on reasoning tasks**.
> >
> > Therefore, we conclude that **performance gains originate from the complex CoT style**.
> >
> > > **W3: Technical Contribution of PTST**
> >
> > We emphasize that **PTST is crucial for training reasoning MLLMs**.
> >
> > PTST was proposed based on the observation that cold-started MLLMs suffer from overthinking, which impedes the optimization process during RL. Therefore, we propose constraining response length in early stages to facilitate learning correct reasoning, then progressively relaxing length constraints to handle more challenging questions. This approach is **critical for mitigating overthinking and achieving optimal performance**. When scaling up to larger models and datasets, this straightforward yet effective method demonstrates consistent strong performance.
> >
> > *As discussed in our response to **W1 of reviewer RiY1***, our method establishes a **standard practice for the domain**. We argue that exploring a simple yet effective solution to a novel problem and validating its effectiveness constitutes a valuable community contribution. Solutions to current challenges need not be complex. When an intuitive approach suffices, subsequent work can build upon our insights to propose more sophisticated methods for further improvements. Therefore, **we believe this does not diminish our contribution**.

---

> > > ### Author Response · Authors · 2025-11-27
> > > **Rebuttal (3/4)**
> > >
> > > > **W4: Further Analysis of Overthinking**
> > >
> > > We thank the reviewer for this suggestion.
> > >
> > > We analyze accuracy versus response length for Vision-R1-CI-7B, Vision-R1-CI-72B, Vision-R1-7B, and Vision-R1-72B on the MathVerse benchmark:
> > >
> > > |                  | 0-4K (proportion/acc.) | 4K-8K (proportion/acc.) | 8K+ (proportion/acc.) |
> > > | ---------------- | ---------------------- | ----------------------- | --------------------- |
> > > | Vision-R1-CI-7B  | 75%/41%                | 11%/28%                 | 14%/7%                |
> > > | Vision-R1-CI-72B | 84%/65%                | 9%/42%                  | 7%/17%                |
> > > | Vision-R1-7B     | 79%/59%                | 18%/31%                 | 3%/18%                |
> > > | Vision-R1-72B    | 98%/65%                | 2%/37%                  | 0%/0%                 |
> > >
> > > Key observations:
> > >
> > > 1. **All models favor shorter responses for correct answers.** We attribute this to **inherent model capacity limitations—merely imitating complex CoT styles makes it difficult to generate long yet correct CoT**. Consequently, excessively long reasoning processes are often erroneous and redundant, leading to the "Overthinking Optimization Problem".
> > >
> > > 2. **Larger models better avoid overthinking.** Comparing Vision-R1-CI-7B and Vision-R1-CI-72B (same cold-start dataset), the 72B model with greater capacity achieves higher accuracy while avoiding excessively redundant ultra-long CoT, completing more correct responses in the shorter 0-4K range.
> > >
> > > 3. **PTST-guided RL effectively mitigates overthinking.** Comparing cold-started models with RL-trained counterparts, PTST-guided learning of correct reasoning patterns enhances reasoning capability and significantly reduces ultra-long CoT generation. Notably, the more capable 72B model, after RL training, fully learns complex yet correct reasoning patterns rather than superficially imitating and producing ultra-long erroneous CoT (producing no 8K+ responses on MathVerse).
> > >
> > > The following table shows accuracy versus length for models cold-started with mulberry-260K on MathVerse:
> > >
> > > |                                  | 0-300 (proportion/acc.) | 300-1K (proportion/acc.) | 1K+ (proportion/acc.) |
> > > | -------------------------------- | ----------------------- | ------------------------ | --------------------- |
> > > | Vision-R1-CI-7B w/ mulberry-260k | 60%/54%                 | 39%/42%                  | 1%/6%                 |
> > >
> > > While "Pseudo-CoT" cold-start produces relatively shorter reasoning processes, **overthinking tendencies still exist**, albeit less pronounced than models cold-started with complex "human-like" Vision-R1-cold dataset. This further demonstrates that **learning complex CoT is more challenging**, but *as shown in **our response to W2***, learning such complex reasoning processes yields substantial reasoning capability improvements far exceeding "Pseudo-CoT".
> > >
> > > > **W5: Potential Limitations of Modality Bridging Pipeline**
> > >
> > > We acknowledge the reviewer's concern regarding potential limitations of Modality Bridging.
> > >
> > > We sampled 100 captions generated during Vision-R1-cold dataset construction and found **only 6 instances** where captions contained partial answer or reasoning information (typically in perception-centric data). Upon further inspection, the generated CoT processes still conform to DeepSeek-R1's complex reasoning style.
> > >
> > > Furthermore, we provide a comparison of data retention rates during VQA data synthesis (after complete post-processing):
> > >
> > > |                                              | Data Sample Size |
> > > | -------------------------------------------- | ---------------- |
> > > | Original data (LLaVA-CoT-100K+Mulberry-260K) | 360K             |
> > > | Vision-R1-cold dataset w/o Modality Bridging | 110K (31%)       |
> > > | **Vision-R1-cold dataset (ours)**            | **200K (56%)**   |
> > >
> > > As shown, Modality Bridging **significantly improves cold-start data retention rates**. Without Modality Bridging, we observe substantial data loss for chart and OCR-centric samples during synthesis, as critical visual details needed for reasoning are missing.
> > >
> > > Our objective with Modality Bridging is to more effectively obtain complex CoT from DeepSeek-R1. While a small fraction of visual descriptions may contain answer-related information, **extensive experiments validate the effectiveness of data obtained through Modality Bridging**, demonstrating its successful application for cold-starting models of varying sizes.

---

> > > > ### Author Response · Authors · 2025-11-27
> > > > **Rebuttal (4/4)**
> > > >
> > > > > **Reference**
> > > >
> > > > [1] Llava-cot: Let vision language models reason step-by-step. ICCV 2025.
> > > >
> > > > [2] Mulberry: Empowering mllm with o1-like reasoning and reflection via collective monte carlo tree search. NeurIPS 2025.
> > > >
> > > > [3] Mmmu: A massive multi-discipline multimodal understanding and reasoning benchmark for expert agi. CVPR 2024.
> > > >
> > > > [4] MMMU-Pro: A More Robust Multi-discipline Multimodal Understanding Benchmark. ACL 2025.
> > > >
> > > > [5] Can mllms reason in multimodality? emma: An enhanced multimodal reasoning benchmark. ICML 2025.
> > > >
> > > > [6] LogicVista: Multimodal LLM Logical Reasoning Benchmark in Visual Contexts. arXiv2407.

---

### Official Review · Reviewer_RiY1 · 2025-10-29

**Soundness:** 3
**Presentation:** 3
**Contribution:** 2
**Rating:** 4
**Confidence:** 5

**Summary:**

This paper introduces Vision-R1, a multimodal large language model (MLLM) designed to enhance reasoning capabilities through reinforcement learning (RL). The model addresses key challenges in incentivizing complex reasoning processes in MLLMs, particularly when dealing with multimodal tasks. Vision-R1 utilizes a novel cold-start initialization approach and Progressive Thinking Suppression Training (PTST) to improve its reasoning abilities. The model outperforms other MLLMs in multimodal math reasoning benchmarks, even with smaller model sizes.

**Strengths:**

Vision-R1 shows competitive results on challenging multimodal benchmarks, with a notable improvement in performance compared to existing models.

**Weaknesses:**

1. Novelty:
While the idea of combining cold-start initialization with RL in MLLMs is presented as a novel approach, it lacks sufficient justification for why this method is fundamentally different from existing methods. The literature already includes several studies that employ RL for reasoning in LLMs (e.g., DeepSeek-R1, Qwen2.5-VL), and this paper does not convincingly demonstrate that the proposed approach significantly advances the field beyond these existing methods. The claim of novelty feels overstated given the incremental nature of the proposed solution.

2. Lack of Comparative Depth:
Although the paper compares Vision-R1 with several state-of-the-art (SoTA) models, the comparison lacks depth in terms of experimental conditions and detailed analyses. For instance, the paper does not provide comprehensive insights into the limitations of the existing approaches it claims to improve upon. A more detailed ablation study that explicitly examines the trade-offs between different design choices (e.g., cold-start initialization vs. direct RL) would strengthen the claims.

3. Evaluation Metrics:
The performance improvement reported (such as a 6% average improvement) is promising but not compelling enough. The results are not sufficiently backed by real-world applications or a broader range of benchmarks that would demonstrate the practical utility of Vision-R1 in diverse settings. There should be more emphasis on generalizability and long-term robustness, particularly in multimodal settings.

4. Over-Emphasis on Technical Complexity:
The method includes complex steps like Progressive Thinking Suppression and Modality Bridging, which are essential for improving performance. However, the paper does not sufficiently explore the potential practical difficulties in implementing these methods at scale. A clearer discussion of how these methods can be generalized or optimized for deployment in real-world scenarios is missing.

**Questions:**

Refer to Weakness.

---

> ### Author Response · Authors · 2025-11-27
> **Rebuttal (1/3)**
>
> > **W1-1: Prior Work on Applying Reinforcement Learning to LLMs**
>
> We thank the reviewer for this valuable feedback. However, we respectfully argue that our contribution differs fundamentally from prior work.
>
> Our contribution is not proposing a novel reinforcement learning algorithm, but rather exploring **how to effectively adapt existing techniques to MLLMs and address domain-specific challenges**. We provide substantial insights, including:
>
> - DeepSeek-R1 achieved remarkable success in incentivizing reasoning capability in LLMs, can this success transfer to the MLLM domain?
> - Directly applying RL to MLLMs does not replicate DeepSeek-R1's success. Given the insufficient base model capabilities and limited data in the multimodal domain, what prevents effective reasoning emergence? Is it the lack of complex "human-like" CoT?
> - If we need to inject complex reasoning generation capabilities involving reflection and questioning into MLLMs, how should we construct effective cold-start data?
> - After cold-start, MLLMs exhibit complex reasoning but suffer from performance degradation and overthinking phenomena that hinder subsequent optimization, how can we effectively elicit reasoning capability during the RL phase?
>
> **These questions for reasoning MLLMs were unexplored and unanswered prior to our work**, and we provide effective solutions. This parallels the contribution of LLaVA [1]: when instruction tuning was successfully applied to LLMs, the MLLM domain still faced unique challenges. Exploring and solving domain-specific problems constitutes significant contribution. **Effectively leveraging existing techniques does not imply lack of novelty**, we have made substantial contributions to multimodal reasoning and established a standard practice for this domain. Therefore, we firmly believe our work significantly advances the field and represents **far more than an "incremental contribution"**.
>
> > **W1-2: Summary of Our Contributions**
>
> We summarize our key contributions as follows:
>
> - **First work to explore how to effectively incentivize complex "human-like" reasoning capability in MLLMs**, identifying and addressing challenges in building complex reasoning MLLMs, thereby establishing a standard practice for the reasoning MLLM domain.
> - **A novel pipeline for constructing high-quality multimodal CoT datasets without human annotation**, producing the 200K Vision-R1-cold dataset with complex CoT for model cold-start. We demonstrate its effectiveness across different model sizes (7B/32B/72B) and data scales, validating it as a high-quality, scalable data synthesis methodology and dataset.
> - **Identification of the overthinking problem in cold-started MLLMs** and proposal of a simple yet effective RL training strategy with progressive thinking suppression that gradually relaxes reasoning length constraints. This approach proves highly effective for scaling up scenarios.

---

> > ### Author Response · Authors · 2025-11-27
> > **Rebuttal (2/3)**
> >
> > > **W2-1: More Detailed Ablation Studies**
> >
> > We thank the reviewer for this suggestion.
> >
> > We provide a comparative experiment using "Pseudo-CoT" to demonstrate limitations of existing approaches. We cold-start Qwen2.5-VL-7B with mulberry-260k [2] and conduct experiments under identical conditions as Vision-R1. *Detailed experimental settings are provided in our response to **W2 of reviewer XezA***. Results are shown below, where "Vision-R1-CI w/ mulberry-260k" and "Vision-R1 w/ mulberry-260k" denote the cold-started model and the RL-trained model, respectively:
> >
> > |                               | MathVista | MathVerse | MM-Math | Avg.        |
> > | ----------------------------- | --------- | --------- | ------- | ----------- |
> > | Qwen-2.5-VL-7B                | 68.1      | 46.7      | 34.1    | 49.6        |
> > | Vision-R1-CI w/ mulberry-260k | 64.4      | 48.5      | 20.1    | 44.3 (-5.3) |
> > | Vision-R1-CI                  | 68.8      | 39.2      | 25.6    | 44.5 (-5.1) |
> > | Vision-R1 w/ mulberry-260k    | 68.4      | 51.6      | 25.6    | 48.5 (-1.1) |
> > | **Vision-R1 (ours)**          | **73.5**  | **52.4**  | **40.2**| **55.4 (+5.8)** |
> >
> > As demonstrated, existing methods such as mulberry utilize step-level CoT to construct the mulberry-260K dataset. When used for cold-start followed by RL training, the reasoning processes learned from step-level "Pseudo-CoT" **fail to achieve effective performance improvements**. We also present the training and validation set rewards during RL (using identical training settings as Vision-R1):
> >
> > |                            | Training Set Reward (Acc.) | Val Set Reward (Acc.) |
> > | -------------------------- | -------------------------- | --------------------- |
> > | Vision-R1 w/ mulberry-260k | 0.64                       | 0.48                  |
> > | Vision-R1 (ours)           | 0.61                       | 0.44                  |
> >
> > Notably, Vision-R1 w/ mulberry-260k achieves higher accuracy rewards on both training and validation sets compared to Vision-R1. However, **due to its inability to generate reasoning processes with complex reflection, it fails to generalize to performance improvements on reasoning tasks**.
> >
> > > **W2-2: Trade-offs Between Different Design Choices**
> >
> > Regarding cold-start initialization versus direct RL, we believe the primary trade-off involves **performance versus inference efficiency**. As shown in Table 3, direct RL produces relatively shorter CoT compared to cold-start initialization (1285 tokens vs. 2057 tokens), yielding higher inference efficiency. However, this length difference primarily stems from direct RL's inability to elicit complex thinking with reflection in MLLMs. Consequently, direct RL (Vision-R1-Zero) fails to achieve performance comparable to cold-start initialization (Vision-R1): 50.7 vs. 55.4. *As discussed in our response to **W2-1***, this complex thinking capability is crucial for model reasoning performance.
> >
> > > **W3-1: Performance Gains Not Compelling Enough**
> >
> > First, we emphasize that **a 6% average improvement on mathematical benchmarks is substantial**. As shown in Table 1, Qwen2.5-VL-72B scales up parameters by 10× compared to Qwen2.5-VL-7B, yet only achieves approximately 8% average performance improvement. Moreover, Qwen2.5-VL is a strong MLLM baseline—improving its performance is non-trivial given the relatively scarce high-quality multimodal data. *As shown in **our response to W2-1***, applying the existing mulberry-260K to Qwen2.5-VL-7B significantly degrades its performance. Furthermore, we conducted experiments on Qwen2.5-VL-32B/72B, achieving **over 10% average performance improvements**. Therefore, we believe our performance gains are sufficiently significant.

---

> > > ### Author Response · Authors · 2025-11-27
> > > **Rebuttal (3/3)**
> > >
> > > > **W3-2: Model Generalization Capability**
> > >
> > > We thank the reviewer for this suggestion.
> > >
> > > We provide additional evaluation results on non-mathematical benchmarks including MMMU (val) [3], MMMU-Pro [4], EMMA [5], and LogicVista [6]:
> > >
> > > |                         | MMMU (val)      | MMMU-Pro        | EMMA            | LogicVista      |
> > > | ----------------------- | --------------- | --------------- | --------------- | --------------- |
> > > | Qwen2.5-VL-7B           | 55.2            | 37.0            | 24.9            | 47.9            |
> > > | **Vision-R1-7B (ours)** | **55.7 (+0.5)** | **37.6 (+0.6)** | **28.2 (+3.3)** | **49.0 (+1.1)** |
> > >
> > > As demonstrated, **Vision-R1 exhibits improved generalization capabilities despite being trained exclusively on mathematical datasets**. While the performance gains on general tasks are less pronounced compared to those on mathematical benchmarks, we attribute this to the training data distribution. We hypothesize that incorporating greater task diversity and volume during Vision-R1 training would yield more substantial improvements. Nonetheless, these results effectively validate that **the human-like complex CoT reasoning introduced by Vision-R1 can effectively enhance the reasoning capabilities of MLLMs**, and importantly, this capability generalizes across domains.
> > >
> > > > **W4: Technical Complexity**
> > >
> > > We thank the reviewer for this valuable feedback.
> > >
> > > **We emphasize that both Progressive Thinking Suppression (PTST) and Modality Bridging are essential components of our approach**.
> > >
> > > PTST was proposed based on the observation that cold-started MLLMs suffer from overthinking, which impedes the optimization process during RL. Therefore, we propose constraining response length in early stages to facilitate learning correct reasoning, then progressively relaxing length constraints to handle more challenging questions. This approach is **critical for mitigating overthinking and achieving optimal performance**.
> > >
> > > For Modality Bridging, we provide a comparison of data retention rates during VQA data synthesis for cold-start datasets (after complete post-processing):
> > >
> > > |                                              | Data Sample Size |
> > > | -------------------------------------------- | ---------------- |
> > > | Original data (LLaVA-CoT-100K+Mulberry-260K) | 360K             |
> > > | Vision-R1-cold dataset w/o Modality Bridging | 110K (31%)       |
> > > | **Vision-R1-cold dataset (ours)**            | **200K (56%)**   |
> > >
> > > As shown, Modality Bridging **significantly improves cold-start data retention rates**. Furthermore, without Modality Bridging, we observe substantial data loss for chart and OCR-centric samples during synthesis, as critical visual details needed for reasoning are missing.
> > >
> > > Additionally, as demonstrated in Table 1, **we conducted experiments scaling up to larger models (32B/72B) and larger datasets, validating that our approach effectively applies to large-scale practical scenarios**.
> > >
> > > Therefore, we believe all proposed components are well-motivated and have been validated in large-scale applications.
> > >
> > > > **Reference**
> > >
> > > [1] Visual instruction tuning. NeurIPS 2023.
> > >
> > > [2] Mulberry: Empowering mllm with o1-like reasoning and reflection via collective monte carlo tree search. NeurIPS 2025.

---

> > > > ### Comment · Reviewer_RiY1 · 2025-11-28
> > > >
> > > > Thanks for the author's response. It has addressed my concerns, and I am willing to raise my score. However, it seems that I am unable to edit my review and score at this moment.

---

> > > > > ### Author Response · Authors · 2025-11-28
> > > > >
> > > > > Dear Reviewer RiY1,
> > > > >
> > > > > Thank you again for your invaluable time and effort in reviewing our paper. We are delighted to note that your concerns have been addressed and that you have raised our score. We deeply appreciate your recognition of our work!
> > > > >
> > > > > Sincerely yours,
> > > > >
> > > > > The Authors

---

### Official Review · Reviewer_i3eD · 2025-11-03

**Soundness:** 3
**Presentation:** 3
**Contribution:** 4
**Rating:** 8
**Confidence:** 3

**Summary:**

The paper introduces a new reasoning model Vision-R1. The authors recognize that RL alone struggles to activate complex reasoning in MLLMs due to insufficient multimodal data, so they propose a novel approach using Progressive Thinking Suppression Training along with RL to refine reasoning processes gradually.The paper also introduces a new 200K multimodal CoT dataset without human annotations to serve as the Cold-start Initialization of Vision-R1 models. Experiments shows Vision-R1 gain better results than previous models with similar size.

**Strengths:**

1. This paper proposes a novel way to generate large dataset without human annotations, which is possible to become a scalable method for dataset creation.
2. Sufficient ablation studies furtther demonstrate how the training parameters in PTST are chosen, and the experiments results shows the effectiveness of propoesed dataset and training strategy.

**Weaknesses:**

1. All the experiments are carried out on math-related dataset (e.g., MathVista, MathVerse). Cross-domain dataset could be included to show the model's power of generalization.
2. While effective, the cold-start process might be limited by the quality and scale of the initial dataset.

**Questions:**

1. Could the model be tested on additional reasoning tasks on other domains to assess its generalization capabilities? For example, MMMU and MMMU-pro dataset.
2. Will the two-stage PTST increase the overall computational overheads? Some statistics about training devices and time may demonstrate the training process more clearly.
3. In tab.5, a three stage PTST layout is included, but it adds a middle stage between original stage 1 and 2. I'd like to know whether adding an additional stage after origin stage 2, like 16K*4, can improve the model's final performance?

---

> ### Author Response · Authors · 2025-11-27
> **Rebuttal (1/2)**
>
> > **W1: Generalization Capability Validation**
>
> We thank the reviewer for this valuable suggestion.
>
> We evaluate our Vision-R1 on multiple non-mathematical multimodal benchmarks, including MMMU (val) [1], MMMU-Pro [2], EMMA [3], and LogicVista [4]. The results are presented in the following table:
>
> |                         | MMMU (val)      | MMMU-Pro        | EMMA            | LogicVista      |
> | ----------------------- | --------------- | --------------- | --------------- | --------------- |
> | Qwen2.5-VL-7B           | 55.2            | 37.0            | 24.9            | 47.9            |
> | **Vision-R1-7B (ours)** | **55.7 (+0.5)** | **37.6 (+0.6)** | **28.2 (+3.3)** | **49.0 (+1.1)** |
>
> As demonstrated, **Vision-R1 exhibits improved generalization capabilities despite being trained exclusively on mathematical datasets**. While the performance gains on general tasks are less pronounced compared to those on mathematical benchmarks, we attribute this to the training data distribution. We hypothesize that incorporating greater task diversity and volume during Vision-R1 training would yield more substantial improvements. Nonetheless, these results effectively validate that **the human-like complex CoT reasoning introduced by Vision-R1 can effectively enhance the reasoning capabilities of MLLMs**, and importantly, this capability generalizes across domains.
>
> > **W2: Limitations of Cold-Start Dataset on Model Performance**
>
> We acknowledge the reviewer's valid concern.
>
> Data quality and scale indeed influence final model performance. However, our results demonstrate that even with relatively limited cold-start data, learning complex natural cognitive processes of questioning and self-reflection combined with reinforcement learning to gradually refine reasoning processes can effectively enhance reasoning capabilities. **This improvement is not only effective at small scales**. As shown in Table 1, when initialized with our proposed 200K cold-start dataset and subsequently scaled up to larger model sizes (32B/72B) with additional RL training data, the reasoning benefits from complex CoT remain consistently present.
>
> Furthermore, *as discussed in **our response to W1***, we believe that future work can achieve even better results by improving the quality and scale of the cold-start dataset using our proposed methodology. Nevertheless, **our current cold-start dataset has been empirically validated to effectively incentivize reasoning capability in multimodal large language models**.
>
> > **Q1: Evaluating Vision-R1 on Other Domain Reasoning Tasks (e.g., MMMU and MMMU-Pro)**
>
> We thank the reviewer for this suggestion.
>
> *Please refer to **our response to W1***, where we provide results on MMMU (val), MMMU-Pro, EMMA, and LogicVista, demonstrating that Vision-R1's reasoning capabilities generalize to non-mathematical domains despite being trained exclusively on mathematical datasets.
>
> > **Q2: Potential Computational Overhead of PTST**
>
> We apologize for not providing training overhead information for PTST in the original submission.
>
> Compared to standard training without PTST, **our approach introduces no additional computational overhead**. While PTST employs different response lengths and sample numbers across stages, we maintain a consistent maximum token budget (e.g., the maximum tokens per prompt rollout is constrained to 64K tokens in both Stage 1 "4K×16" and Stage 2 "8K×8"). This design ensures that the average rollout overhead remains approximately constant across different PTST stages. The following table reports the average wall-clock time per training step for Vision-R1-7B using 8 nodes with 8×H800 GPUs each:
>
> |                             | Stage 1 (4K×16) | Stage 2 (8K×8) | Stage 3 (16K×4) |
> | --------------------------- | --------------- | -------------- | --------------- |
> | Avg. training time per step | ~680s           | ~640s          | ~670s           |

---

> > ### Author Response · Authors · 2025-11-27
> > **Rebuttal (2/2)**
> >
> > > **Q3: PTST Stage 3 Performance**
> >
> > We thank the reviewer for this suggestion.
> >
> > We provide additional performance comparisons for Stage 3 in the table below. The corresponding training curves can be found in Figure 1(E) of the manuscript.
> >
> > | Method                              | Stage 1   | Stage 2  | Stage 3   | MathVista | MathVerse | MM-Math | Avg.  |
> > | ----------------------------------- | --------- | -------- | --------- | --------- | --------- | ------- | ----- |
> > | Vision-R1                           | 4K×16     | 8K×8     | −         | 73.5      | 52.4      | 40.2    | 55.4  |
> > | **Vision-R1 w/ additional Stage 3** | **4K×16** | **8K×8** | **16K×4** | **72.3**  | **50.2**  | **37.6**| **53.4** |
> >
> > As shown, incorporating an additional Stage 3 does not improve model performance. We hypothesize that **this phenomenon stems from inherent model capacity limitations in handling extremely long reasoning chains** (*as also discussed in our response to **W4 of reviewer XezA***). Although our progressive training strategy initially constrains response length to facilitate learning of correct reasoning processes and subsequently relaxes this constraint, the benefits of extended reasoning remain bounded by the model's intrinsic reasoning capacity (excessively long reasoning may introduce redundancy beyond the model's effective utilization). In our experiments, we find that an 8K length constraint represents a reasonable trade-off for Vision-R1's reasoning capability.
> >
> >
> > > **Reference**
> >
> > [1] Mmmu: A massive multi-discipline multimodal understanding and reasoning benchmark for expert agi. CVPR 2024.
> >
> > [2] MMMU-Pro: A More Robust Multi-discipline Multimodal Understanding Benchmark. ACL 2025.
> >
> > [3] Can mllms reason in multimodality? emma: An enhanced multimodal reasoning benchmark. ICML 2025.
> >
> > [4] LogicVista: Multimodal LLM Logical Reasoning Benchmark in Visual Contexts. arXiv2407.

---

> > > ### Comment · Reviewer_i3eD · 2025-11-28
> > >
> > > Thanks for your detailed response. The response addressed most of my concerns and proofed the generalization ability of the proposed method.

---

> > > > ### Author Response · Authors · 2025-11-28
> > > >
> > > > Dear Reviewer i3eD,
> > > >
> > > > Thank you again for your invaluable time and effort on our paper. Thank you very much for approving our work!
> > > >
> > > > Sincerely Yours,
> > > >
> > > > The Authors

---

### Meta-Review · Area_Chair_1epb · 2025-12-27

**Summary:**

The paper proposes Vision-R1, a framework to enhance the reasoning capability of Multimodal Large Language Models (MLLMs) using Reinforcement Learning (RL). Inspired by DeepSeek-R1-Zero’s success in LLMs, Vision-R1 addresses the key challenge of activating complex reasoning (e.g., questioning, reflection) in MLLMs—an issue rooted in the scarcity of high-quality multimodal reasoning data. The solution involves three core components: (1) constructing a 200K human-annotation-free multimodal Chain-of-Thought (CoT) dataset (Vision-R1-cold) via modality bridging and filtering, using existing MLLMs and DeepSeek-R1 for cold-start initialization; (2) introducing a Progressive Thinking Suppression Training (PTST) strategy to mitigate post-cold-start overthinking; and (3) employing Group Relative Policy Optimization (GRPO) with a hard formatting result reward function to refine the model’s ability to learn correct, complex reasoning processes. Empirically, Vision-R1 achieves notable results: Vision-R1-7B reaches 73.5% accuracy on MathVista (only 0.4% below OpenAI O1), averages ~6% improvement across multimodal math benchmarks with just 10K RL training data, and scales effectively (Vision-R1-32B: 76.4%, Vision-R1-72B: 78.2% on MathVista).

Reviewers’ core concerns informing the decision included:
* Generalization beyond math tasks (Reviewer i3eD).
* Novelty vs. prior RL (Reviewer RiY1, Lj3M).
* In-depth analysis of each proposed component (Reviewer XezA).
* PTST mechanism clarity and insufficient ablation study (Reviewer 4piz).

Reviewers originally have scores (8,4,4,4) and raise to (8,6,6,4, considering the potential score increase from the reviewers’ reply), since most of the concerns are addressed post-rebuttal. Since the merits of this borderline paper outweigh the flaws, I recommend acceptance.

**Reviewer Concerns:**

The concerns from Reviewer XezA, such as further analysis on each component, such as the Pseudo-CoT, Progressive Thinking Suppression Training, Overthinking Optimization Problem, and the "Modality Bridging" pipeline, might still be outstanding.

**Reviewer Scores:**

Reviewer RiY1 and Reviewer Lj3M promised to increase their scores post-rebuttal, since the authors' response addresses most of their concerns.

---

### Decision · Program_Chairs · 2026-01-26

Accept (Poster)